# ADAVI: Automatic Dual Amortized Variational Inference Applied To Pyramidal Bayesian Models

**Louis Rouillard**
Université Paris-Saclay, Inria, CEA
Palaiseau, 91120, France
`louis.rouillard-odera@inria.fr`

**Demian Wassermann**
Université Paris-Saclay, Inria, CEA
Palaiseau, 91120, France
`demian.wassermann@inria.fr`

## Abstract

Frequently, population studies feature pyramidally-organized data represented using Hierarchical Bayesian Models (HBM) enriched with plates. These models can become prohibitively large in settings such as neuroimaging, where a sample is composed of a functional MRI signal measured on 300 brain locations, across 4 measurement sessions, and 30 subjects, resulting in around 1 million latent parameters. Such high dimensionality hampers the usage of modern, expressive flow-based techniques. To infer parameter posterior distributions in this challenging class of problems, we designed a novel methodology that automatically produces a variational family dual to a target HBM. This variational family, represented as a neural network, consists in the combination of an attention-based hierarchical encoder feeding summary statistics to a set of normalizing flows. Our automatically-derived neural network exploits exchangeability in the plate-enriched HBM and factorizes its parameter space. The resulting architecture reduces by orders of magnitude its parameterization with respect to that of a typical flow-based representation, while maintaining expressivity. Our method performs inference on the specified HBM in an amortized setup: once trained, it can readily be applied to a new data sample to compute the parameters' full posterior. We demonstrate the capability and scalability of our method on simulated data, as well as a challenging high-dimensional brain parcellation experiment. We also open up several questions that lie at the intersection between normalizing flows, SBI, structured Variational Inference, and inference amortization.

## 1 Introduction

Inference aims at obtaining the posterior distribution $p(\theta|X)$ of latent model parameters $\theta$ given the observed data $X$. In the context of Hierarchical Bayesian Models (HBM), $p(\theta|X)$ usually has no known analytical form, and can be of a complex shape -different from the prior's (Gelman et al., 2004). Modern normalizing-flows based techniques -universal density estimators- can overcome this difficulty (Papamakarios et al., 2019a; Ambrogioni et al., 2021b). Yet, in setups such as neuroimaging, featuring HBMs representing large population studies (Kong et al., 2019; Bonkhoff et al., 2021), the dimensionality of $\theta$ can go over the million. This high dimensionality hinders the usage of normalizing flows, since their parameterization usually scales quadratically with the size of the parameter space (e.g. Dinh et al., 2017; Papamakarios et al., 2018; Grathwohl et al., 2018). Population studies with large dimensional features are therefore inaccessible to off-the-shelf flow-based techniques and their superior expressivity. This can in turn lead to complex, problem-specific derivations: for instance Kong et al. (2019) rely on a manually-derived Expectation Maximization (EM) technique. Such an analytical complexity constitutes a strong barrier to entry, and limits the wide and fruitful usage of Bayesian modelling in fields such as neuroimaging. Our main aim is to meet that experimental need: how can we derive a technique both automatic and efficient in the context of very large, hierarchically-organised data?

Approximate inference features a large corpus of methods including Monte Carlo methods (Koller & Friedman, 2009) and Variational Auto Encoders (Zhang et al., 2019). We take particular inspiration

Figure 1: Automatic Dual Amortized Variational Inference (ADAVI) working principle. On the left is a generative HBM, with 2 alternative representations: a graph *template* featuring 2 plates $\mathcal{P}_0$, $\mathcal{P}_1$ of cardinality 2, and the equivalent *ground* graph depicting a typical pyramidal shape. We note $\mathcal{B}_1 = \mathrm{Card}(\mathcal{P}_1)$ the batch shape due to the cardinality of $\mathcal{P}_1$. The model features 3 latent RV $\lambda$, $\kappa$ and $\mathbf{\Gamma} = [\gamma^1, \gamma^2]$, and one observed RV $\mathbf{X} = [[x^{1,1}, x^{1,2}], [x^{2,1}, x^{2,2}]]$. We analyse automatically the structure of the HBM to produce its *dual* amortized variational family (on the right). The hierarchical encoder HE processes the observed data $X$ through 2 successive set transformers ST to produce encodings $\mathbf{E}$ aggregating summary statistics at different hierarchies. Those encodings are then used to condition density estimators -the combination of a normalizing flow $\mathcal{F}$ and a link function $l$- producing the variational distributions for each latent RV.

from the field of Variational Inference (VI) (Blei et al., 2017), deemed to be most adapted to large parameter spaces. In VI, the experimenter posits a variational family $\mathcal{Q}$ so as to approximate $q(\theta) \approx p(\theta|X)$. In practice, deriving an expressive, yet computationally attractive variational family can be challenging (Blei et al., 2017). This triggered a trend towards the derivation of automatic VI techniques (Kucukelbir et al., 2016; Ranganath et al., 2013; Ambrogioni et al., 2021b). We follow that logic and present a methodology that automatically derives a variational family $\mathcal{Q}$. In Fig. 1, from the HBM on the left we derive automatically a neural network architecture on the right. We aim at deriving our variational family $\mathcal{Q}$ in the context of *amortized* inference (Rezende & Mohamed, 2016; Cranmer et al., 2020). Amortization is usually obtained at the cost of an *amortization gap* from the true posterior, that accumulates on top of a *approximation gap* dependent on the expressivity of the variational family $\mathcal{Q}$ (Cremer et al., 2018). However, once an initial training overhead has been "paid for", amortization means that our technique can be applied to a any number of data points to perform inference in a few seconds.

Due to the very large parameter spaces presented above, our target applications aren't amenable to the generic flow-based techniques described in Cranmer et al. (2020) or Ambrogioni et al. (2021b). We therefore differentiate ourselves in exploiting the invariance of the problem not only through the design of an adapted encoder, but down to the very architecture of our density estimator. Specifically, we focus on the inference problem for Hierarchical Bayesian Models (HBMs) (Gelman et al., 2004; Rodrigues et al., 2021). The idea to condition the architecture of a density estimator by an analysis of the dependency structure of an HBM has been studied in (Wehenkel & Louppe, 2020; Weilbach et al., 2020), in the form of the masking of a single normalizing flow. With Ambrogioni et al. (2021b), we instead share the idea to combine multiple separate flows. More generally, our static analysis of a generative model can be associated with structured VI (Hoffman & Blei, 2014; Ambrogioni et al., 2021a;b). Yet our working principles are rather orthogonal: structured VI usually aims at exploiting model structure to augment the expressivity of a variational family, whereas we aim at reducing its parameterization.

Our objective is therefore to derive an automatic methodology that takes as input a generative HBM and generates a *dual* variational family able to perform amortized parameter inference. This variational family exploits the exchangeability in the HBM to reduce its parameterization by orders of magnitude compared to generic methods (Papamakarios et al., 2019b; Greenberg et al., 2019;

Ambrogioni et al., 2021b). Consequently, our method can be applied in the context of large, pyramidally-structured data, a challenging setup inaccessible to existing flow-based methods and their superior expressivity. We apply our method to such a large pyramidal setup in the context of neuroimaging (section 3.5), but demonstrate the benefit of our method beyond that scope. Our general scheme is visible in Fig. 1, a figure that we will explain throughout the course of the next section.

## 2 METHODS

### 2.1 PYRAMIDAL BAYESIAN MODELS

We are interested in experimental setups modelled using plate-enriched Hierarchical Bayesian Models (HBMs) (Kong et al., 2019; Bonkhoff et al., 2021). These models feature independent sampling from a common conditional distribution at multiple levels, translating the graphical notion of *plates* (Gilks et al., 1994). This nested structure, combined with large measurements -such as the ones in fMRI- can result in massive latent parameter spaces. For instance the population study in Kong et al. (2019) features multiple subjects, with multiple measures per subject, and multiple brain vertices per measure, for a latent space of around $0.4$ million parameters. Our method aims at performing inference in the context of those large plate-enriched HBMs.

Such HBMs can be represented with Directed Acyclic Graphs (DAG) templates (Koller & Friedman, 2009) with vertices -corresponding to RVs- $\{\theta_i\}_{i=0...L}$ and plates $\{\mathcal{P}_p\}_{p=0...P}$. We denote as $\mathrm{Card}(\mathcal{P})$ the -fixed- *cardinality* of the plate $\mathcal{P}$, i.e. the number of independent draws from a common conditional distribution it corresponds to. In a template DAG, a given RV $\theta$ can belong to multiple plates $\mathcal{P}_h, \ldots \mathcal{P}_P$. When *grounding* the template DAG into a ground graph -instantiating the repeated structure symbolized by the plates $\mathcal{P}$- $\theta$ would correspond to multiple RVs of similar parametric form $\{\theta^{i_h,\ldots,i_P}\}$, with $i_h = 1 \ldots \mathrm{Card}(\mathcal{P}_h)$, $\ldots$, $i_P = 1 \ldots \mathrm{Card}(\mathcal{P}_P)$. This equivalence visible on the left on Fig. 1, where the template RV $\mathbf{\Gamma}$ corresponds to the ground RVs $[\gamma^1, \gamma^2]$. We wish to exploit this plate-induced exchangeability.

We define the sub-class of models we specialize upon as *pyramidal* models, which are plate-enriched DAG templates with the 2 following differentiating properties. First, we consider a single stack of the plates $\mathcal{P}_0, \ldots, \mathcal{P}_P$. This means that any RV $\theta$ belonging to plate $\mathcal{P}_p$ also belongs to plates $\{\mathcal{P}_q\}_{q>p}$. We thus don't treat in this work the case of *colliding* plates (Koller & Friedman, 2009). Second, we consider a single observed RV $\theta_0$, with observed value $X$, belonging to the plate $\mathcal{P}_0$ (with no other -latent- RV belonging to $\mathcal{P}_0$). The obtained graph follows a typical pyramidal structure, with the observed RV at the basis of the pyramid, as seen in Fig. 1. This figure features 2 plates $\mathcal{P}_0$ and $\mathcal{P}_1$, the observed RV is $\mathbf{X}$, at the basis of the pyramid, and latent RVs are $\mathbf{\Gamma}$, $\lambda$ and $\kappa$ at upper levels of the pyramid. Pyramidal HBMs delineate models that typically arise as part of population studies -for instance in neuroimaging- featuring a nested group structure and data observed at the subject level only (Kong et al., 2019; Bonkhoff et al., 2021).

The fact that we consider a single pyramid of plates allows us to define the *hierarchy* of an RV $\theta_i$ denoted $\mathrm{Hier}(\theta_i)$. An RV's *hierarchy* is the level of the pyramid it is placed at. Due to our pyramidal structure, the observed RV will systematically be at hierarchy 0 and latent RVs at hierarchies $> 0$. For instance, in the example in Fig. 1 the observed RV $\mathbf{X}$ is at hierarchy 0, $\mathbf{\Gamma}$ is at hierarchy 1 and both $\lambda$ and $\kappa$ are at hierarchy 2.

Our methodology is designed to process generative models whose dependency structure follows a pyramidal graph, and to scale favorably when the plate cardinality in such models augments. Given the observed data $X$, we wish to obtain the posterior density for latent parameters $\theta_1, \ldots, \theta_L$, exploiting the exchangeability induced by the plates $\mathcal{P}_0, \ldots, \mathcal{P}_P$.

### 2.2 AUTOMATIC DERIVATION OF A DUAL AMORTIZED VARIATIONAL FAMILY

In this section, we derive our main methodological contribution. We aim at obtaining posterior distributions for a generative model of pyramidal structure. For this purpose, we construct a family of variational distributions $\mathcal{Q}$ *dual* to the model. This architecture consists in the combination of 2 items. First, a Hierarchical Encoder (HE) that aggregates summary statistics from the data. Second, a set of conditional density estimators.

**Tensor functions** We first introduce the notations for tensor functions which we define in the spirit of Magnus & Neudecker (1999). We leverage tensor functions throughout our entire architecture to reduce its parameterization. Consider a function $f : F \to G$, and a tensor $\mathbf{T}_F \in F^{\mathcal{B}}$ of shape $\mathcal{B}$. We denote the tensor $\mathbf{T}_G \in G^{\mathcal{B}}$ resulting from the element-wise application of $f$ over $\mathbf{T}_F$ as $\mathbf{T}_G = \overrightarrow{f}^{(\mathcal{B})}(\mathbf{T}_F)$ (in reference to the programming notion of *vectorization* in Harris et al. (2020)). In Fig. 1, $\overrightarrow{\mathrm{ST}_0}^{(\mathcal{B}_1)}$ and $\overrightarrow{l_\gamma \circ \mathcal{F}_\gamma}^{(\mathcal{B}_1)}$ are examples of tensor functions. At multiple points in our architecture, we will translate the repeated structure in the HBM induced by plates into the repeated usage of functions across plates.

**Hierarchical Encoder** For our encoder, our goal is to learn a function HE that takes as input the observed data $X$ and successively exploits the permutation invariance across plates $\mathcal{P}_0, \ldots, \mathcal{P}_P$. In doing so, HE produces encodings $\mathbf{E}$ at different hierarchy levels. Through those encodings, our goal is to learn summary statistics from the observed data, that will condition our amortized inference. For instance in Fig. 1, the application of HE over $X$ produces the encodings $\mathbf{E}_1$ and $\mathbf{E}_2$.

To build HE, we need at multiple hierarchies to collect summary statistics across i.i.d samples from a common distribution. To this end we leverage *SetTransformers* (Lee et al., 2019): an attention-based, permutation-invariant architecture. We use *SetTransformers* to derive encodings across a given plate, repeating their usage for all larger-rank plates. We cast the observed data $X$ as the encoding $\mathbf{E}_0$. Then, recursively for every hierarchy $h = 1 \ldots P + 1$, we define the encoding $\mathbf{E}_h$ as the application to the encoding $\mathbf{E}_{h-1}$ of the tensor function corresponding to the set transformer $\mathrm{ST}_{h-1}$. $\mathrm{HE}(X)$ then corresponds to the set of encodings $\{\mathbf{E}_1, \ldots, \mathbf{E}_{P+1}\}$ obtained from the successive application of $\{\mathrm{ST}_h\}_{h=0,\ldots,P}$. If we denote the batch shape $\mathcal{B}_h = \mathrm{Card}(\mathcal{P}_h) \times \ldots \times \mathrm{Card}(\mathcal{P}_P)$:

$$\mathbf{E}_h = \overrightarrow{\mathrm{ST}}_{h-1}^{(\mathcal{B}_h)}(\mathbf{E}_{h-1}) \quad \mathrm{HE}(X) = \{\mathbf{E}_1, \ldots, \mathbf{E}_{P+1}\} \tag{1}$$

In collecting summary statistics across the i.i.d. samples in plate $\mathcal{P}_{h-1}$, we decrease the order of the encoding tensor $\mathbf{E}_{h-1}$. We *repeat* this operation in parallel on every plate of larger rank than the rank of the contracted plate. We consequently produce an encoding tensor $\mathbf{E}_h$ with the batch shape $\mathcal{B}_h$, which is the batch shape of every RV of hierarchy $h$. In that line, successively summarizing plates $\mathcal{P}_0, \ldots, \mathcal{P}_P$, of increasing rank results in encoding tensors $\mathbf{E}_1, \ldots, \mathbf{E}_{P+1}$ of decreasing order. In Fig. 1, there are 2 plates $\mathcal{P}_0$ and $\mathcal{P}_1$, hence 2 encodings $\mathbf{E}_1 = \overrightarrow{\mathrm{ST}_0}^{(\mathcal{B}_1)}(X)$ and $\mathbf{E}_2 = \mathrm{ST}_1(\mathbf{E}_1)$. $\mathbf{E}_1$ is an order 2 tensor: it has a batch shape of $\mathcal{B}_1 = \mathrm{Card}(\mathcal{P}_1)$ -similar to $\mathbf{\Gamma}$- whereas $\mathbf{E}_2$ is an order 1 tensor. We can decompose $\mathbf{E}_1 = [e_1^1, e_1^2] = [\mathrm{ST}_0([X^{1,1}, X^{1,2}]), \mathrm{ST}_0([X^{2,1}, X^{2,2}])]$.

**Conditional density estimators** We now will use the encodings $\mathbf{E}$, gathering hierarchical summary statistics on the data $X$, to condition the inference on the parameters $\theta$. The encodings $\{\mathbf{E}_h\}_{h=1\ldots P+1}$ will respectively condition the density estimators for the posterior distribution of parameters sharing their hierarchy $\{\{\theta_i : \mathrm{Hier}(\theta_i) = h\}\}_{h=1\ldots P+1}$.

Consider a latent RV $\theta_i$ of hierarchy $h_i = \mathrm{Hier}(\theta_i)$. Due to the plate structure of the graph, $\theta_i$ can be decomposed in a batch of shape $\mathcal{B}_{h_i} = \mathrm{Card}(\mathcal{P}_{h_i}) \times \ldots \times \mathrm{Card}(\mathcal{P}_P)$ of multiple similar, conditionally independent RVs of individual size $S_{\theta_i}$. This decomposition is akin to the grounding of the considered graph template (Koller & Friedman, 2009). A conditional density estimator is a 2-step diffeomorphism from a latent space onto the event space in which the RV $\theta_i$ lives. We initially parameterize every variational density as a standard normal distribution in the latent space $\mathbb{R}^{S_{\theta_i}}$. First, this latent distribution is reparameterized by a conditional *normalizing flow* $\mathcal{F}_i$ (Rezende & Mohamed, 2016; Papamakarios et al., 2019a) into a distribution of more complex density in the space $\mathbb{R}^{S_{\theta_i}}$. The flow $\mathcal{F}_i$ is a diffeomorphism in the space $\mathbb{R}^{S_{\theta_i}}$ conditioned by the encoding $\mathbf{E}_{h_i}$. Second, the obtained latent distribution is projected onto the event space in which $\theta_i$ lives by the application of a *link function* diffeomorphism $l_i$. For instance, if $\theta_i$ is a variance parameter, the link function would map $\mathbb{R}$ onto $\mathbb{R}^{+*}$ ($l_i = \mathrm{Exp}$ as an example). The usage of $\mathcal{F}_i$ and the link function $l_i$ is *repeated* on plates of larger rank than the hierarchy $h_i$ of $\theta_i$. The resulting conditional density estimator $q_i$ for the posterior distribution $p(\theta_i|X)$ is given by:

$$u_i \sim \mathcal{N}\left(\overrightarrow{0}_{\mathcal{B}_{h_i} \times S_{\theta_i}}, \boldsymbol{I}_{\mathcal{B}_{h_i} \times S_{\theta_i}}\right) \quad \tilde{\theta}_i = \overrightarrow{l_i \circ \mathcal{F}_i}^{(\mathcal{B}_{h_i})}(u_i; \mathbf{E}_{h_i}) \sim q_i(\theta_i; \mathbf{E}_{h_i}) \tag{2}$$

In Fig. 1 $\mathbf{\Gamma} = [\gamma^1, \gamma^2]$ is associated to the diffeomorphism $\overrightarrow{l_\gamma \circ \mathcal{F}_\gamma}^{(\mathcal{B}_1)}$. This diffeomorphism is conditioned by the encoding $\mathbf{E}_1$. Both $\mathbf{\Gamma}$ and $\mathbf{E}_1$ share the batch shape $\mathcal{B}_1 = \mathrm{Card}(\mathcal{P}_1)$. Decomposing the encoding $\mathbf{E}_1 = [e_1^1, e_1^2]$, $e_1^1$ is used to condition the inference on $\gamma^1$, and $e_1^2$ for $\gamma^2$. $\lambda$ is associated to the diffeomorphism $l_\lambda \circ \mathcal{F}_\lambda$, and $\kappa$ to $l_\kappa \circ \mathcal{F}_\kappa$, both conditioned by $\mathbf{E}_2$.

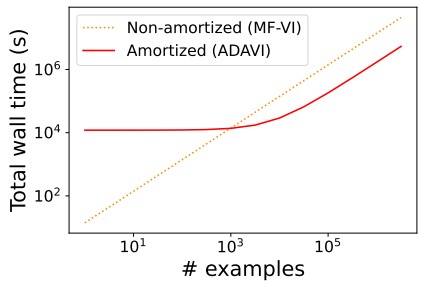

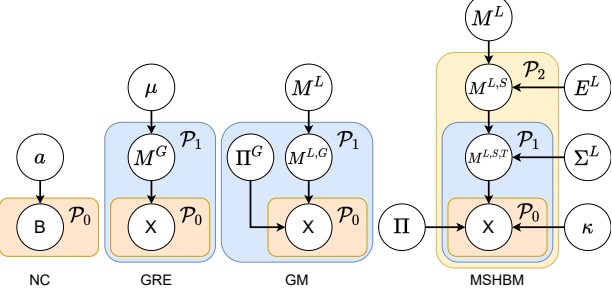

(a) Cumulative GPU training and inference time for a non-amortized (MF-VI) and an amortized (ADAVI) method.

(b) Experiment's HBMs from left to right: Non-conjugate (NC) (section 3.2), Gaussian random effects (GRE) (3.1, 3.3), Gaussian mixture (GM) (3.4), Multi-scale (MSHBM) (3.5).

Figure 2: **panel (a)**: inference amortization on the Gaussian random effects example defined in eq. (6): as the number of examples rises, the amortized method becomes more attractive; **panel (b)**: graph templates corresponding to the HBMs presented as part of our experiments.

**Parsimonious parameterization** Our approach produces a parameterization effectively independent from plate cardinalities. Consider the latent RVs $\theta_1, \dots, \theta_L$. Normalizing flow-based density estimators have a parameterization quadratic with respect to the size of the space they are applied to (e.g. Papamakarios et al., 2018). Applying a single normalizing flow to the total event space of $\theta_1, \dots, \theta_L$ would thus result in $\mathcal{O}([\sum_{i=1}^{L} S_{\theta_i} \prod_{p=h_i}^{P} \text{Card}(\mathcal{P}_p)]^2)$ weights. But since we instead apply multiple flows on the spaces of size $S_{\theta_i}$ and repeat their usage across all plates $\mathcal{P}_{h_i}, \dots, \mathcal{P}_P$, we effectively reduce this parameterization to:

$$\# \text{ weights}_{\text{ADAVI}} = \mathcal{O}\left(\sum_{i=1}^{L} S_{\theta_i}^2\right) \tag{3}$$

As a consequence, our method can be applied to HBMs featuring large plate cardinalities without scaling up its parameterization to impractical ranges, preventing a computer memory blow-up.

## 2.3 VARIATIONAL DISTRIBUTION AND TRAINING

Given the encodings $\mathbf{E}_p$ provided by HE, and the conditioned density estimators $q_i$, we define our parametric amortized variational distribution as a *mean field approximation* (Blei et al., 2017):

$$q_{\chi, \Phi}(\theta|X) = q_\Phi(\theta; \text{HE}_\chi(X)) = \prod_{i=1 \dots L} q_i(\theta_i; \mathbf{E}_{h_i}, \Phi) \tag{4}$$

In Fig. 1, we factorize $q(\mathbf{\Gamma}, \kappa, \lambda|X) = q_\gamma(\mathbf{\Gamma}; \mathbf{E}_1) \times q_\lambda(\lambda; \mathbf{E}_2) \times q_\kappa(\kappa; \mathbf{E}_2)$. Grouping parameters as $\Psi = (\chi, \Phi)$, our objective is to have $q_\Psi(\theta|X) \approx p(\theta|X)$. Our loss is an amortized version of the classical ELBO expression (Blei et al., 2017; Rezende & Mohamed, 2016):

$$\Psi^\star = \arg\min_\Psi \frac{1}{M} \sum_{m=1}^{M} \log q_\Psi(\theta^m|X^m) - \log p(X^m, \theta^m), \quad X^m \sim p(X), \theta^m \sim q_\Psi(\theta|X) \tag{5}$$

Where we denote $z \sim p(z)$ the sampling of $z$ according to the distribution p(z). We jointly train HE and $q_i, i = 1 \dots L$ to minimize the amortized ELBO. The resulting architecture performs amortized inference on latent parameters. Furthermore, since our parameterization is invariant to plate cardinalities, our architecture is suited for population studies with large-dimensional feature space.

## 3 EXPERIMENTS

In the following experiments, we consider a variety of inference problems on pyramidal HBMs. We first illustrate the notion of amortization (section 3.1). We then test the expressivity (section 3.2, 3.4), scalability (section 3.3) of our architecture, as well as its practicality on a challenging neuroimaging experiment (section 3.5).

**Baseline choice** In our experiments we use as baselines: *Mean Field VI* (MF-VI) (Blei et al., 2017) is a common-practice method; *(Sequential) Neural Posterior Estimation* (NPE-C, SNPE-C) (Greenberg et al., 2019) is a structure-unaware, *likelihood-free* method: SNPE-C results from the sequential -and no longer amortized- usage of NPE-C; *Total Latent Space Flow* (TLSF) (Rezende & Mohamed, 2016) is a reverse-KL counterpoint to SNPE-C: both fit a single normalizing flow to the entirety of the latent parameter space but SNPE-C uses a forward KL loss while TLSF uses a reverse KL loss; *Cascading Flows* (CF) (Ambrogioni et al., 2021b) is a structure-aware, prior-aware method: CF-A is our main point of comparison in this section. For relevant methods, the suffix -(N)A designates the (non) amortized implementation. More details related to the choice and implementation of those baselines can be found in our supplemental material.

## 3.1 INFERENCE AMORTIZATION

In this experiment we illustrate the trade-off between amortized versus non-amortized techniques (Cranmer et al., 2020). For this, we define the following Gaussian random effects HBM (Gelman et al., 2004) (see Fig. 2b-GRE):

$$D, \ N = 2, \ 50 \qquad \qquad \mu \sim \mathcal{N}(\vec{0}_D, \sigma_\mu^2)$$
$$G = 3 \qquad \qquad \mu^g | \mu \sim \mathcal{N}(\mu, \sigma_g^2) \qquad M^G = [\mu^g]^{g=1...G} \qquad (6)$$
$$\sigma_\mu, \ \sigma_g, \ \sigma_x = 1.0, \ 0.2, \ 0.05 \quad x^{g,n} | \mu^g \sim \mathcal{N}(\mu^g, \sigma_x^2) \qquad X = [x^{g,n}]_{n=1...N}^{g=1...G}$$

In Fig. 2a we compare the cumulative time to perform inference upon a batch of examples drawn from this generative HBM. For a single example, a non-amortized technique can be faster -and deliver a posterior closer to the ground truth- than an amortized technique. This is because the non-amortized technique fits a solution for this specific example, and can tune it extensively. In terms of ELBO, on top of an *approximation gap* an amortized technique will add an *amortization gap* (Cremer et al., 2018). On the other hand, when presented with a new example, the amortized technique can infer directly whereas the optimization of the non-amortized technique has to be repeated. As the number of examples rises, an amortized technique becomes more and more attractive. This result puts in perspective the quantitative comparison later on performed between amortized and non-amortized techniques, that are qualitatively distinct.

## 3.2 EXPRESSIVITY IN A NON-CONJUGATE CASE

In this experiment, we underline the superior expressivity gained from using normalizing flows - used by ADAVI or CF- instead of distributions of fixed parametric form -used by MF-VI. For this we consider the following HBM (see Fig. 2b-NC):

$$N, \ D = 10, \ 2 \qquad \qquad r_a, \ \sigma_b = 0.5, \ 0.3$$
$$a \sim Gamma(\vec{1}_D, r_a) \qquad b^n | a \sim Laplace(a, \sigma_b) \quad B = [b^n]^{n=1...N} \qquad (7)$$

This example is voluntarily non-canonical: we place ourselves in a setup where the posterior distribution of $a$ given an observed value from $B$ has no known parametric form, and in particular is not of the same parametric form as the prior. Such an example is called *non-conjugate* in Gelman et al. (2004). Results are visible in table 1-NC: MF-VI is limited in its ability to approximate the correct distribution as it attempts to fit to the posterior a distribution of the same parametric form as the prior. As a consequence, contrary to the experiments in section 3.3 and section 3.4 -where MF-VI stands as a strong baseline- here both ADAVI and CF-A are able to surpass its performance.

**Proxy to the ground truth posterior** MF-VI plays the role of an ELBO upper bound in our experiments GRE (section 3.3), GM (section 3.4) and MS-HBM (section 3.5). We crafted those examples to be conjugate: MF-VI thus doesn't feature any approximation gap, meaning $\mathrm{KL}(q(\theta)||p(\theta|X)) \simeq 0$. As such, its $ELBO(q) = \log p(X) - \mathrm{KL}(q(\theta)||p(\theta|X))$ is approximately equal to the evidence of the observed data. As a consequence, any inference method with the same ELBO value -calculated over the same examples- as MF-VI would yield an approximate posterior with low KL divergence to the true posterior. Our main focus in this work are flow-based methods, whose performance would be maintained in non-conjugate cases, contrary to MF-VI (Papamakarios et al., 2019a). We further focus on amortized methods, providing faster inference for a multiplicity of problem instances, see e.g. section 3.1. MF-VI is therefore not to be taken as part of a benchmark but as a proxy to the unknown ground truth posterior.

Table 1: Expressivity comparison on the non-conjugate (NC) and Gaussian mixture (GM) examples. **NC**: both CF-A and ADAVI show higher ELBO than MF-VI. **GM**: TLSF-A and ADAVI show higher ELBO than CF-A, but do not reach the ELBO levels of MF-VI, TLSF-NA and CF-NA. Are compared from left to right: ELBO median (larger is better) and standard deviation; for non-amortized techniques: CPU inference time for one example (seconds); for amortized techniques: CPU amortization time (seconds). Methods are ran over 20 random seeds, except for SNPE-C and TLSF-NA who were ran on 5 seeds per sample, for a number of effective runs of 100. For CF, the ELBO designates the numerically comparable *augmented ELBO* (Ranganath et al., 2016).

| HBM | Type | Method | ELBO | Inf. (s) | Amo. (s) |
|---|---|---|---|---|---|
| NC | Fixed param. form | MF-VI | -21.0 ($\pm$ 0.2) | 17 | - |
| (section 3.2) | Flow-based | CF-A | -17.5 ($\pm$ 0.1) | - | 220 |
| | | **ADAVI** | -17.6 ($\pm$ 0.3) | - | 1,000 |
| GM | Ground truth proxy | MF-VI | 171 ($\pm$ 970) | 23 | - |
| (section 3.4) | Non amortized | SNPE-C | -14,800 ($\pm$ 15,000) | 70,000 | - |
| | | TLSF-NA | 181 ($\pm$ 680) | 330 | - |
| | | CF-NA | 191 ($\pm$ 390) | 240 | - |
| | Amortized | NPE-C | -27,000 ($\pm$ 19,000) | - | 1300 |
| | | TLSF-A | -530 ($\pm$ 980) | - | 360,000 |
| | | CF-A | -7,000 ($\pm$ 640) | - | 23,000 |
| | | **ADAVI** | -494 ($\pm$ 430) | - | 150,000 |

### 3.3 PERFORMANCE SCALING WITH RESPECT TO PLATE CARDINALITY

In this experiment, we illustrate our plate cardinality independent parameterization defined in section 2.2. We consider 3 instances of the Gaussian random effects model presented in eq. (6), increasing the number of groups from $G = 3$ to $G = 30$ and $G = 300$. In doing so, we augment the total size of the latent parametric space from 8 to 62 to 602 parameters, and the observed data size from 300 to 3,000 to 30,000 values. Results for this experiment are visible in Fig. 3 (see also table 2). On this example we note that amortized techniques only feature a small amortization gap (Cremer et al., 2018), reaching the performance of non-amortized techniques -as measured by the ELBO, using MF-VI as an upper bound- at the cost of large amortization times. We note that the performance of (S)NPE-C quickly degrades as the plate dimensionality augments, while TLSF's performance is maintained, hinting towards the advantages of using the likelihood function when available. As the HBM's plate cardinality augments, we match the performance and amortization time of state-of-the-art methods, but we do so maintaining a constant parameterization.

### 3.4 EXPRESSIVITY IN A CHALLENGING SETUP

In this experiment, we test our architecture on a challenging setup in inference: a mixture model. Mixture models notably suffer from the *label switching* issue and from a loss landscape with multiple strong local minima (Jasra et al., 2005). We consider the following mixture HBM (see Fig. 2b-GM):

$$\kappa, \ \sigma_\mu, \ \sigma_g, \ \sigma_x = 1, \ 1.0, \ 0.2, \ 0.05 \qquad G, L, D, N = 3, 3, 2, 50$$

$$\mu_l \sim \mathcal{N}(\vec{0}_D, \sigma_\mu^2) \qquad\qquad M^L = [\mu_l]^{l=1...L} \qquad\qquad (8a)$$

$$\mu_l^g | \mu_l \sim \mathcal{N}(\mu_l, \sigma_g^2) \qquad\qquad M^{L,G} = [\mu_l^g]_{g=1...G}^{l=1...L}$$

$$\pi^g \in [0,1]^L \sim \text{Dir}([\kappa] \times L) \qquad\qquad \Pi^G = [\pi^g]^{g=1...G}$$

$$x^{g,n} | \pi^g, [\mu_1^g, \ldots, \mu_L^g] \sim \text{Mix}(\pi^g, [\mathcal{N}(\mu_1^g, \sigma_x^2) \ldots \mathcal{N}(\mu_L^g, \sigma_x^2)]) \qquad X = [x^{g,n}]_{n=1...N}^{g=1...G} \qquad (8b)$$

Where $\text{Mix}(\pi, [p_1, \ldots, p_N])$ denotes the finite mixture of the densities $[p_1, \ldots, p_N]$ with $\pi$ the mixture weights. The results are visible in table 1-GM. In this complex example, similar to TLSF-A we obtain significantly higher ELBO than CF-A, but we do feature an amortization gap, not reaching the ELBO level of non-amortized techniques. We also note that despite our efforts (S)NPE-C failed to

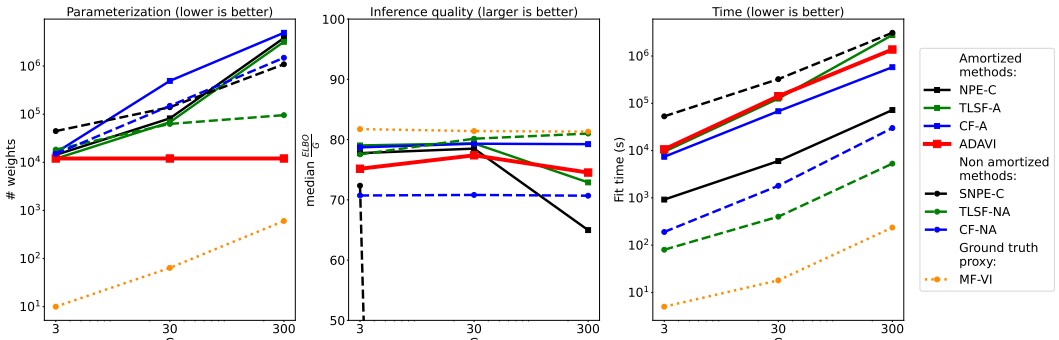

Figure 3: Scaling comparison on the Gaussian random effects example. ADAVI -in red- maintains constant parameterization as the plates cardinality goes up (first panel); it does so while maintaining its inference quality (second panel) and a comparable amortization time (third panel). Are compared from left to right: number of weights in the model; closeness of the approximate posterior to the ground truth via the $\frac{ELBO}{G}$ median -that allows for a comparable numerical range as G augments; CPU amortization + inference time (s) for a single example -this metric advantages non-amortized methods. *Non-amortized* techniques are represented using dashed lines, and *amortized* techniques using plain lines. *MF-VI*, in dotted lines, plays the role of the upper bound for the ELBO. Results for SNPE-C and NPE-C have to be put in perspective, as from $G = 30$ and $G = 300$ respectively both methods reach data regimes in which the inference quality is very degraded (see table 2). Implementation details are shared with table 1.

reach the ELBO level of other techniques. We interpret this result as the consequence of a forward-KL-based training taking the full blunt of the *label switching* problem, as seen in appendix D.2. Fig. D.3 shows how our higher ELBO translates into results of greater experimental value.

### 3.5  NEUROIMAGING: MODELLING MULTI-SCALE VARIABILITY IN BROCA'S AREA FUNCTIONAL PARCELLATION

To show the practicality of our method in a high-dimensional context, we consider the model proposed by Kong et al. (2019). We apply this HBM to parcel the human brain's Inferior Frontal Gyrus in 2 functional MRI (fMRI)-based connectivity networks. Data is extracted from the Human Connectome Project dataset (Van Essen et al., 2012). The HBM models a population study with 30 subjects and 4 large fMRI measures per subject, as seen in Fig. 2b-MSHBM: this nested structure creates a large latent space of $\simeq 0.4$ million parameters and an even larger observed data size of $\simeq 50$ million values. Due to our parsimonious parameterization, described in eq. (4), we can nonetheless tackle this parameter range without a memory blow-up, contrary to all other presented flow-based methods -CF, TLSF, NPE-C. Resulting population connectivity profiles can be seen in Fig. 4. We are in addition interested in the stability of the recovered population connectivity considering subsets of the population. For this we are to sample without replacement hundreds of sub-populations of 5 subjects from our population. On GPU, the inference wall time for MF-VI is 160 seconds per sub-population, for a mean $\log(-\text{ELBO})$ of $28.6(\pm0.2)$ (across 20 examples, 5 seeds per example). MF-VI can again be considered as an ELBO upper bound. Indeed the MSHBM can be considered as a 3-level (subject, session, vertex) Gaussian mixture with random effects, and therefore features conjugacy. For multiple sub-populations, the total inference time for MF-VI reaches several hours. On the contrary, ADAVI is an amortized technique, and as such features an amortization time of 550 seconds, after which it can infer on any number of sub-populations in a few seconds. The posterior quality is similar: a mean $\log(-\text{ELBO})$ of $29.0(\pm0.01)$. As shown in our supplemental material -as a more meaningful comparison- the resulting difference in the downstream parcellation task is marginal (Fig. E.7).

We therefore bring the expressivity of flow-based methods and the speed of amortized techniques to parameter ranges previously unreachable. This is due to our plate cardinality-independent parameterization. What's more, our automatic method only necessitates a practitioner to declare the generative HBM, therefore reducing the analytical barrier to entry there exists in fields such as neu-

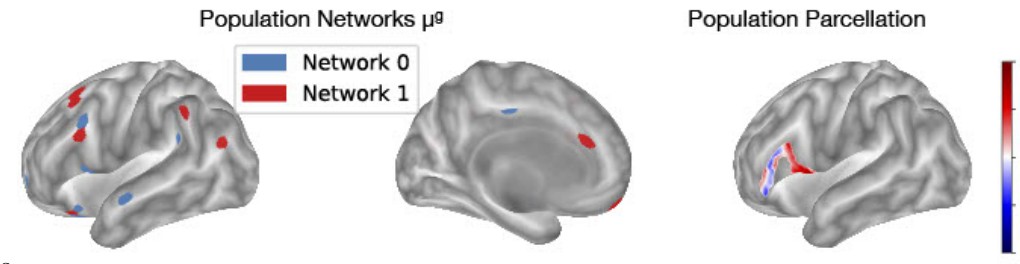

s

Figure 4: Results for our neuroimaging experiment. On the left, networks show the top 1% connected components. Network 0 (in blue) agrees with current knowledge in semantic/phonologic processing while network 1 (in red) agrees with current networks known in language production (Heim et al., 2009; Zhang et al., 2020). Our soft parcellation, where coloring lightens as the cortical point is less probably associated with one of the networks, also agrees with current knowledge where more posterior parts are involved in language production while more anterior ones in semantic/phonological processing (Heim et al., 2009; Zhang et al., 2020).

roimaging for large-scale Bayesian analysis. Details about this experiment, along with subject-level results, can be found in our supplemental material.

## 4 DISCUSSION

**Exploiting structure in inference** In the SBI and VAE setups, data structure can be exploited through learnable data embedders (Zhang et al., 2019; Radev et al., 2020). We go one step beyond and also use the problem structure to shape our density estimator: we factorize the parameter space of a problem into smaller components, and share network parameterization across tasks we know to be equivalent (see section 2.2 and 3.3). In essence, we construct our architecture not based on a ground HBM graph, but onto its template, a principle that could be generalized to other types of templates, such as temporal models (Koller & Friedman, 2009). Contrary to the notion of black box, we argue that experimenters oftentimes can identify properties such as exchangeability in their experiments (Gelman et al., 2004). As our experiments illustrate (section 3.4, section 3.3), there is much value in exploiting this structure. Beyond the sole notion of plates, a static analysis of a forward model could automatically identify other desirable properties that could be then leveraged for efficient inference. This concept points towards fruitful connections to be made with the field of *lifted* inference (Broeck et al., 2021; Chen et al., 2020).

**Mean-Field approximation** A limitation in our work is that our posterior distribution is akin to a mean field approximation (Blei et al., 2017): with the current design, no statistical dependencies can be modelled between the RV blocks over which we fit normalizing flows (see section 2.3). Regrouping RV *templates*, we could model more dependencies at a given hierarchy. On the contrary, our method prevents the direct modelling of dependencies between *ground* RVs corresponding to repeated instances of the same template. Those dependencies can arise as part of inference (Webb et al., 2018). We made the choice of the Mean Field approximation to streamline our contribution, and allow for a clear delineation of the advantages of our methods, not tying them up to a method augmenting a variational family with statistical dependencies, an open research subject (Ambrogioni et al., 2021b; Weilbach et al., 2020). Though computationally attractive, the mean field approximation nonetheless limits the expressivity of our variational family (Ranganath et al., 2016; Hoffman & Blei, 2014). We ponder the possibility to leverage VI architectures such as the one derived by Ranganath et al. (2016); Ambrogioni et al. (2021b) and their augmented variational objectives for structured populations of normalizing flows such as ours.

**Conclusion** For the delineated yet expressive class of pyramidal Bayesian models, we have introduced a potent, automatically derived architecture able to perform amortized parameter inference. Through a Hierarchical Encoder, our method conditions a network of normalizing flows that stands as a variational family *dual* to the forward HBM. To demonstrate the expressivity and scalability of our method, we successfully applied it to a challenging neuroimaging setup. Our work stands as an original attempt to leverage exchangeability in a generative model.

ACKNOWLEDGMENTS

This work was supported by the ERC-StG NeuroLang ID:757672.

We would like to warmly thank Dr. Thomas Yeo and Dr. Ru Kong (CBIG) who made pre-processed HCP functional connectivity data available to us.

We also would like to thank Dr. Majd Abdallah (Inria) for his insights and perspectives regarding our functional connectivity results.

REPRODUCIBILITY STATEMENT

All experiments were performed on a computational cluster with 16 Intel(R) Xeon(R) CPU E5-2660 v2 @ 2.20GHz (256Mb RAM), 16 AMD EPYC 7742 64-Core Processor (512Mb RAM) CPUs and 1 NVIDIA Quadro RTX 6000 (22Gb), 1 Tesla V100 (32Gb) GPUs.

All methods were implemented in Python. We implemented most methods using *Tensorflow Probability* (Dillon et al., 2017), and SBI methods using the SBI Python library (Tejero-Cantero et al., 2020).

As part of our submission we release the code associated to our experiments. Our supplemental material furthermore contains an entire section dedicated to the implementation details of the baseline methods presented as part of our experiments. For our neuromimaging experiment, we also provide a section dedicated to our pre-processing and post-processing steps

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

This supplemental material complements our main work both with theoretical points and experiments:

A complements to our methods section 2. We present the HBM descriptors needed for the automatic derivation of our dual architecture;

B complements to our discussion section 3. We elaborate on various points including amortization;

C complements to the Gaussian random effects experiment described in eq. (6). We present results mostly related to hyperparameter analysis;

D complements to the Gaussian mixture with random effects experiment (section 3.4). We explore the complexity of the example at hand;

E complements to the MS-HBM experiments (section 3.5). We present some context for the experiment, a toy dimensions experiment and implementation details.

F justification and implementation details for the baseline architectures used in our experiments;

## A COMPLEMENTS TO THE METHODS: MODEL DESCRIPTORS FOR AUTOMATIC VARIATIONAL FAMILY DERIVATION

This section is a complement to section 2. We formalize explicitly the *descriptors* of the generative HBM needed for our method to derive its dual architecture. This information is of experimental value, since those descriptors need to be available in any API designed to implement our method.

If we denote $\mathrm{plates}(\theta)$ the plates the RV $\theta$ belongs to, then the following HBM descriptors are the needed input to derive automatically our ADAVI dual architecture:

$$
\begin{aligned}
\mathcal{V} &= \{\theta_i\}_{i=0\dots L} \\
\mathrm{P} &= \{\mathcal{P}_p\}_{p=0\dots P} \\
\mathrm{Card} &= \{\mathcal{P}_p \to \#\mathcal{P}_p\}_{p=0\dots P} \\
\mathrm{Hier} &= \{\theta_i \mapsto h_i = \min_p\{p \,:\, \mathcal{P}_p \in \mathrm{plates}(\theta_i)\}\}_{i=0\dots L} \\
\mathrm{Shape} &= \{\theta_i \mapsto \mathcal{S}^{\mathrm{event}}_{\theta_i}\}_{i=0\dots L} \\
\mathrm{Link} &= \{\theta_i \mapsto (l_i : S_{\theta_i} \to S^{\mathrm{event}}_{\theta_i})\}_{i=0\dots L}
\end{aligned}
\tag{A.1}
$$

Where:

- $\mathcal{V}$ lists the RVs in the HBM (vertices in the HBM's corresponding graph template);
- $\mathrm{P}$ lists the plates in the HBM's graph template;
- $\mathrm{Card}$ maps a plate $\mathcal{P}$ to its *cardinality*, that is to say the number of independent draws from a common conditional density it corresponds to;
- $\mathrm{Hier}$ maps a RV $\theta$ to its *hierarchy*, that is to say the level of the pyramid it is placed at, or equivalently the smallest rank for the plates it belongs to;
- $\mathrm{Shape}$ maps a RV to its *event shape* $S^{\mathrm{event}}_{\theta_i}$. Consider the plate-enriched graph template representing the HBM. A single graph template RV belonging to plates corresponds to multiple similar RVs when grounding this graph template. $S^{\mathrm{event}}_{\theta_i}$ is the potentially high-order shape for any of those multiple ground RVs.
- $\mathrm{Link}$ maps a RV $\theta$ to its *link function* $l$. The *Link function* projects the latent space for the RV $\theta$ onto the event space in which $\theta$ lives. For instance, if $\theta$ is a variance parameter, the link function would map $\mathbb{R}$ onto $\mathbb{R}^{+*}$ ($l = \mathrm{Exp}$ as an example). Note that the latent space of shape $S_\theta$ is necessary an order 1 unbounded real space. $l$ therefore potentially implies a reshaping to the high-order shape $S^{\mathrm{event}}_{\theta_i}$.

Those descriptors can be readily obtained from a static analysis of a generative model, especially when the latter is expressed in a modern probabilistic programming framework (Dillon et al., 2017; Bingham et al., 2019).

# B   COMPLEMENTS TO OUR DISCUSSION

## B.1   AMORTIZATION

Contrary to traditional VI, we aim at deriving a variational family $\mathcal{Q}$ in the context of amortized inference (Rezende & Mohamed, 2016; Cranmer et al., 2020). This means that, once an initial training overhead has been "paid for", our technique can readily be applied to a new data point. Amortized inference is an active area of research in the context of Variational Auto Encoders (VAE) (Kingma & Welling, 2014; Wu et al., 2019; Shu et al., 2018; Iakovleva et al., 2020). It is also a the original setup of normalizing flows (NF) (Rezende & Mohamed, 2016; Radev et al., 2020), our technology of choice. From this amortized starting point, Cranmer et al. (2020); Papamakarios et al. (2019b); Thomas et al. (2020); Greenberg et al. (2019) have notably developed *sequential* techniques, refining a posterior -and losing amortization- across several rounds of simulation. To streamline our contribution, we chose not build upon that research, and rather focus on the amortized implementation of normalizing flows. But we argue that our contribution is actually rather orthogonal to those: similar to Ambrogioni et al. (2021b) we propose a principled and automated way to combine several density estimators in a hierarchical structure. As such, our methods could be applied to a different class of estimators such as VAEs (Kingma & Welling, 2014). We could leverage the SBI techniques and extend our work into a sequential version through the reparameterization of our conditional estimators $q_i$ (see section 2.2). Ultimately, our method is not meant as an alternative to SBI, but a complement to it for the pyramidal class of problems described in section 2.1.

We choose to posit ourselves as an amortized technique. Yet, in our target experiment from Kong et al. (2019) (see section 3.5), the inference is performed on a specific data point. An amortized method could therefore appear as a more natural option. What's more, it is generally admitted that amortized inference implies an *amortization gap* from the true posterior, which accumulates on top of the *approximation gap* that depends on the expressivity of the considered variational family. This amortization gap further reduces the quality of the approximate posterior for a given data point.

Our experimental experience on the example in section 3.4 however makes us put forth the value that can be obtained from sharing learning across multiple examples, as amortization entitles (Cranmer et al., 2020). Specifically, we encountered less issues related to local minima of the loss, a canonical issue for MF-VI (Blei et al., 2017) that is for instance illustrated in our supplemental material. We would therefore argue against the intuition that a (locally) amortized technique is necessarily wasteful in the context of a single data point. However, as the results in table 2 and table 1 underline, there is much work to be done for amortized technique to reach the performance consistency and training time of amortized techniques, especially in high dimension, where exponentially more training examples can be necessary to estimate densities properly (Donoho, 2000).

Specializing for a local parameter regime -as sequential method entitles (Cranmer et al., 2020)- could therefore make us benefit from amortization without too steep an upfront training cost.

## B.2   EXTENSION TO A BROADER CLASS OF SIMULATORS

The presence of exchangeability in a problem's data structure is not tied to the explicit modelling of a problem as a HBM: Zaheer et al. (2018) rather describe this property as an permutation invariance present in the studied data. As a consequence, though our derivation is based on HBMs, we believe that the working principle of our method could be applied to a broader class of simulators featuring exchangeability. Our reliance on HBMs is in fact only tied to our usage of the reverse KL loss (see section 2.3), a readily modifiable implementation detail.

In this work, we restrict ourselves to the pyramidal class of Bayesian networks (see section 2.1). Going further, this class of models could be extended to cover more and more use-cases. This bottom-up approach stands at opposite ends from the generic approach of SBI techniques (Cranmer et al., 2020). But, as our target experiment in 3.5 demonstrates, we argue that in the long run this bottom-up approach could result in more scalable and efficient architectures, applicable to challenging setups such as neuroimaging.

## B.3 RELEVANCE OF LIKELIHOOD-FREE METHODS IN THE PRESENCE OF A LIKELIHOOD

As part of our benchmark, we made the choice to include likelihood-free methods (NPE-C and SNPE-C), based on a forward KL loss. In our supplemental material (appendix C.4) we also study the implementation of our method using a forward KL loss.

There is a general belief in the research community that likelihood-free methods are not intended to be as competitive as likelihood-based methods in the presence of a likelihood (Cranmer et al., 2020). In this manuscript, we tried to provide quantitative results to nourish this debate. We would argue that likelihood-free methods generally scaled poorly to high dimensions (section 3.3). The result of the Gaussian Mixture experiment also shows poorer performance in a multi-modal case, but we would argue that the performance drop of likelihood-free methods is actually largely due to the label switching problem (see appendix D.2). On the other hand, likelihood-free methods are dramatically faster to train and can perform on par with likelihood-based methods in examples such as the Gaussian Random Effects for $G = 3$ (see table 2). Depending on the problem at hand, it is therefore not straightforward to systematically disregard likelihood-free methods.

As an opening, there maybe is more at the intersection between likelihood-free and likelihood-based methods than meets the eye. The symmetric loss introduced by Weilbach et al. (2020) stands as a fruitful example of that connection.

## B.4 INFERENCE OVER A SUBSET OF THE LATENT PARAMETERS

Depending on the downstream tasks, out of all the parameters $\theta$, an experimenter could only be interested in the inference of a subset $\Theta_1$. Decomposing $\theta = \Theta_1 \cup \Theta_2$, the goal would be to derive a variational distribution $q_1(\Theta_1)$ instead of the distribution $q(\Theta_1, \Theta_2)$.

**Reverse KL setup**    We first consider the reverse KL setup. The original ELBO maximized as part of inference is equal to:

$$\begin{aligned} \text{ELBO}(q) &= \log p(X) - \text{KL}[q(\Theta_1, \Theta_2) || p(\Theta_1, \Theta_2 | X)] \\ &= \mathbb{E}_q[\log p(X, \Theta_1, \Theta_2) - \log q(\Theta_1, \Theta_2)] \end{aligned} \tag{B.2}$$

To keep working with normalized distributions, we get a similar expression for the inference of $\Theta_1$ only via:

$$\text{ELBO}(q_1) = \mathbb{E}_{q_1}[\log p(X, \Theta_1) - \log q_1(\Theta_1)] \tag{B.3}$$

In this expression, $p(X, \Theta_1)$ is unknown: it results from the marginalization of $\Theta_2$ in $p$, which is non-trivial to obtain, even via a Monte Carlo scheme. As a consequence, working with the reverse KL does not allow for the inference over a subset of latent parameters.

**Forward KL setup**    Contrary to reverse KL, in the forward KL setup the evaluation of $p$ is not required. Instead, the variational family is trained using samples $(\theta, X)$ from the joint distribution $p$. In this setup, inference can be directly restricted over the parameter subset $\Theta_1$. Effectively, one wouldn't have to construct density estimators for the parameters $\Theta_2$, and the latter would be marginalized in the obtained variational distribution $q(\Theta_1)$. However, as our experiments point out (section 3.3, section 3.4), likelihood-free training can be less competitive in large data regimes or complex inference problems. As a consequence, even if this permits inference over only the parameters of interest, switching to a forward KL loss can be inconvenient.

## B.5 EMBEDDING SIZE FOR THE HIERARCHICAL ENCODER

An important hyper-parameter in our architecture is the embedding size for the *Set Transformer* (ST) architecture (Lee et al., 2019). The impact of the embedding size for a single ST has already been studied in Lee et al. (2019), as a consequence we didn't devote any experiments to the study of the impact of this hyper-parameter.

However, our architecture stacks multiple ST networks, and the evolution of the embedding size with the hierarchy could be an interesting subject:

- it is our understanding that the embedding size for the encoding $\mathbf{E}_h$ should be increasing with:
  - the number of latent RVs $\theta$ whose inference depends on $\mathbf{E}_h$, i.e. the latent RVs of hierarchy $h$
  - the dimensionality of the latent RVs $\theta$ of hierarchy $h$
  - the complexity of the inference problem at hand, for instance how many statistical moments need to be computed from i.i.d data points

- experimentally, we kept the embedding size constant across hierarchies, and fixed this constant value based on the aforementioned criteria (see appendix F.2). This approach is probably conservative and drives up the number of weights in HE

- higher-hierarchy encodings are constructed from sets of lower-hierarchy encodings. Should the embedding size vary, it would be important not to "bottleneck" the information collected at low hierarchies, even if the aforementioned criteria would argue for a low embedding size.

There would be probably experimental interest in deriving algorithms estimating the optimal embedding size at different hierarchies. We leave this to future work.

## B.6 Bounds for ADAVI's inference performance

When considering an amortized variational family, the non-amortized family with the same parametric form can be considered as an upper bound for the inference performance -as measured by the ELBO. Indeed, considering the fixed parametric family $q(\theta; \Psi)$, for a given data point $X^1$ the best performance can be obtained by freely setting up the $\Psi^1$ parameters. Instead setting $\Psi^1 = f(X^1)$ -amortizing the inference- can only result in worst performance. On the other hand the parameters for another data point $X^2$ can then readily be obtained via $\Psi^2 = f(X^2)$ (Cremer et al., 2018).

In a similar fashion, it can be useful to look for upper bounds for ADAVI's performance. This is notably useful to compare ADAVI to traditional MF-VI (Blei et al., 2017):

1. **Base scenario: traditional MF-VI** In traditional MF-VI, the variational distribution is $q^{\text{MF-VI}} = \prod_i q_i^{\text{Prior's parametric form}}(\theta_i)$:
   - $q_i^{\text{Prior's parametric form}}$ can for instance be a Gaussian with parametric mean and variance;
   - in non-conjugate cases, using the prior's parametric form can result in poor performance due to an approximation gap, as seen in section 3.2;
   - due to the difference in expressivity introduced by normalizing flows, except in conjugate cases, $q^{\text{MF-VI}}$ is **not** an upper bound for ADAVI's performance.

2. **Superior upper limit scenario: normalizing flows using the Mean Field approximation** A family more expressive then $q^{\text{MF-VI}}$ can be obtained via a collection of normalizing flows combined using the mean field approximation: $q^{\text{MF-NF}} = \prod_i q_i^{\text{Normalizing flow}}(\theta_i)$:
   - every individual $q_i^{\text{Normalizing flow}}$ is more expressive than the corresponding $q_i^{\text{Prior's parametric form}}$: in a non-conjugate case it would provide better performance (Papamakarios et al., 2019a);
   - since the mean field approximation treats the inference over each $\theta_i$ as a separate problem, the resulting distribution $q^{\text{MF-NF}}$ is more expressive than $q^{\text{MF-VI}}$;
   - consider a plate-enriched DAG (Koller & Friedman, 2009), a *template* RV $\theta_i$, and $\theta_i^j$ with $j = 1 \ldots \text{Card}(\mathcal{P})$ the corresponding *ground* RVs. In $q^{\text{MF-NF}}$, every $\theta_i^j$ would be associated to a separate normalizing flow;
   - consequently, the parameterization of $q^{\text{MF-NF}}$ is linear with respect to $\text{Card}(\mathcal{P})$. This is less than the quadratic scaling of TLSF or NPE-C -as explained in section 2.2 and appendix F.2. But this scaling still makes $q^{\text{MF-NF}}$ not adapted to large plate cardinalities, all the more since the added number of weights -corresponding to a full normalizing flow- per $\theta_i^j$ is high;

- this scaling is similar to the one of Cascading Flows (Ambrogioni et al., 2021b): CF can be considered as the improvement of $q^{\text{MF-NF}}$ with statistical dependencies between the $q_i$;
- as far as we know, the literature doesn't feature instances of the $q^{\text{MF-NF}}$ architecture. Though straightforward, the introduction of normalizing flows in a variational family is non-trivial, and for instance marks the main difference between CF and its predecessor ASVI (Ambrogioni et al., 2021a).

3. **Inferior upper limit scenario: non-amortized ADAVI** At this point, it is useful to consider the non-existent architecture $q^{\text{ADAVI-NA}}$:

- compared to $q^{\text{MF-NF}}$, considering the *ground* RVs $\theta_i^j$ corresponding to the *template* RV $\theta_i$, each $\theta_i^j$ would no longer correspond to a different normalizing flow, but to the same conditional normalizing flow;
- each $\theta_i^j$ would then be associated to a separate independent encoding vector. There wouldn't be a need for our Hierarchical Encoder anymore -as referenced to in section 2.2;
- as for $q^{\text{MF-NF}}$, the parameterization of $q^{\text{ADAVI-NA}}$ would scale linearly with $\text{Card}(\mathcal{P})$. Each new $\theta_i^j$ would only necessitate an additional embedding vector, which would make $q^{\text{ADAVI-NA}}$ more adapted to high plate cardinalities than $q^{\text{MF-NF}}$ or CF;
- using separate flows for the $\theta_i^j$ instead of a shared conditional flow, $q^{\text{MF-NF}}$ can be considered as an upper bound for $q^{\text{ADAVI-NA}}$'s performance;
- due to the amortization gap, $q^{\text{ADAVI-NA}}$ can be considered as an upper bound for ADAVI's performance. By transitivity, $q^{\text{MF-NF}}$ is then an even higher bound for ADAVI's performance.

It is to be noted that amongst the architectures presented above, ADAVI is the only architecture with a parameterization invariant to the plate cardinalities. This brings the advantage to theoretically being able to use ADAVI on plates of any cardinality, as seen in eq. (4). In that sense, our main claim is tied to the amortization of our variational family, though the linear scaling of $q^{\text{ADAVI-NA}}$ could probably be acceptable for reasonable plate cardinalities.

## C   COMPLEMENTS TO THE GAUSSIAN RANDOM EFFECTS EXPERIMENT: HYPERPARAMETER ANALYSIS

This section features additional results on the experiment described in eq. (6) with $G = 3$ groups. We present results of practical value, mostly related to hyperparameters.

### C.1   DESCRIPTORS, INPUTS TO ADAVI

We can analyse the model described in eq. (6) using the descriptors defined in eq. (A.1). Those descriptors constitute the inputs our methodology needs to automatically derive the *dual* architecture from the generative HBM:

$$
\begin{aligned}
\mathcal{V} &= \{\mu, M^G, X\} \\
\mathrm{P} &= \{\mathcal{P}_0, \mathcal{P}_1\} \\
\mathrm{Card} &= \{\mathcal{P}_0 \mapsto N, \mathcal{P}_1 \mapsto G\} \\
\mathrm{Hier} &= \{\mu \mapsto 2, M^G \mapsto 1, X \mapsto 0\} \\
\mathrm{Shape} &= \{\mu \mapsto (D,), M^G \mapsto (D,), X \mapsto (D,)\} \\
\mathrm{Link} &= \{\mu \mapsto \mathrm{Identity}, M^G \mapsto \mathrm{Identity}, X \mapsto \mathrm{Identity}\}
\end{aligned}
\tag{C.4}
$$

### C.2   TABULAR RESULTS FOR THE SCALING EXPERIMENT

A tabular representation of the results presented in Fig. 3 can be seen in table 2.

| G | Type | Method | ELBO ($10^2$) | # weights | Inf. (s) | Amo. (s) |
|---|---|---|---|---|---|---|
| 3 | Grd truth proxy | MF-VI | 2.45 ($\pm$ 0.15) | 10 | 5 | - |
| | Non amortized | SNPE-C | 2.17 ($\pm$ 33) | 45,000 | 53,000 | - |
| | | TLSF-NA | 2.33 ($\pm$ 0.20) | 18,000 | 80 | - |
| | | CF-NA | 2.12 ($\pm$ 0.15) | 15,000 | 190 | - |
| | Amortized | NPE-C | 2.33 ($\pm$ 0.15) | 12,000 | - | 920 |
| | | TLSF-A | 2.37 ($\pm$ 0.072) | 12,000 | - | 9,400 |
| | | CF-A | 2.36 ($\pm$ 0.029) | 16,000 | - | 7,400 |
| | | **ADAVI** | 2.25 ($\pm$ 0.14) | 12,000 | - | 11,000 |
| 30 | Grd truth proxy | MF-VI | 24.4 ($\pm$ 0.41) | 64 | 18 | - |
| | Non amortized | SNPE-C | -187 ($\pm$ 110) | 140,000 | 320,000 | - |
| | | TLSF-NA | 24.0 ($\pm$ 0.49) | 63,000 | 400 | - |
| | | CF-NA | 21,2 ($\pm$ 0.40) | 150,000 | 1,800 | - |
| | Amortized | NPE-C | 23.6 ($\pm$ 50) | 68,000 | - | 6,000 |
| | | TLSF-A | 22.7 ($\pm$ 13) | 68,000 | - | 130,000 |
| | | CF-A | 23.8 ($\pm$ 0.06) | 490,000 | - | 68,000 |
| | | **ADAVI** | 23.2 ($\pm$ 0.89) | 12,000 | - | 140,000 |
| 300 | Grd truth proxy | MF-VI | 244 ($\pm$ 1.3) | 600 | 240 | - |
| | Non amortized | SNPE-C | -9,630 ($\pm$ 3,500) | 1,100,000 | 3,100,000 | - |
| | | TLSF-NA | 243 ($\pm$ 1.8) | 960,000 | 5,300 | - |
| | | CF-NA | 212 ($\pm$ 1.5) | 1,500,000 | 30,000 | - |
| | Amortized | NPE-C | 195 ($\pm 3 \times 10^6$)[1] | 3,200,000 | - | 72,000 |
| | | TLSF-A | 202 ($\pm$ 120) | 3,200,000 | - | 2,800,000 |
| | | CF-A | 238 ($\pm$ 0.1) | 4,900,000 | - | 580,000 |
| | | **ADAVI** | 224 ($\pm$ 9.4) | 12,000 | - | 1,300,000 |

Table 2: Scaling comparison on the Gaussian random effects example (see Fig. 2b-GRE). Methods are ran over 20 random seeds (Except for SNPE-C and TLSF: to limit computational resources usage, those non-amortized computationally intensive methods were only ran on 5 seeds per sample, for a number of effective runs of 100). Are compared: from left to right ELBO median (higher is better) and standard deviation (ELBO for all techniques except for Cascading Flows, for which ELBO is the numerically comparable *augmented ELBO* (Ranganath et al., 2016)); number of trainable parameters (weights) in the model; for non-amortized techniques: CPU inference time for one example (seconds); for amortized techniques: CPU amortization time (seconds). [1]- Results for NPE-C are extremely unstable, with multiple NaN results: the median value is rather random and not necessarily indicative of a good performance

## C.3 DERIVATION OF AN ANALYTIC POSTERIOR

To have a ground truth to which we can compare our methods results, we derive the following analytic posterior distributions. Assuming we know $\sigma_\mu$, $\sigma_g$, $\sigma_x$:

$$\hat{\mu}^g = \frac{1}{N} \sum_{n=1}^{N} x_n^g \tag{C.5a}$$

$$\tilde{\mu}^g | \hat{\mu}^g \sim \mathcal{N} \left( \hat{\mu}^g, \frac{\sigma_x^2}{N} \operatorname{Id}_D \right) \tag{C.5b}$$

$$\hat{\mu} = \frac{1}{G} \sum_{g=1}^{G} \hat{\mu}^g \tag{C.5c}$$

$$\tilde{\mu} | \hat{\mu} \sim \mathcal{N} \left( \frac{\frac{G}{\sigma_g^2} \hat{\mu}}{\frac{1}{\sigma_\mu^2} + \frac{G}{\sigma_g^2}}, \frac{1}{\frac{1}{\sigma_\mu^2} + \frac{G}{\sigma_g^2}} \operatorname{Id}_D \right) \tag{C.5d}$$

Where in equation C.5b we neglect the influence of the prior (against the evidence) on the posterior in light of the large number of points drawn from the distribution. We note that this analytical posterior is *conjugate*, as argued in section 3.2.

## C.4 TRAINING LOSSES FULL DERIVATION AND COMPARISON

**Full formal derivation** Following the nomenclature introduced in Papamakarios et al. (2019a), there are 2 different ways in which we could train our variational distribution:

- using a *forward* KL divergence, benefiting from the fact that we can sample from our generative model to produce a dataset $\{(\theta^m, X^m)\}_{m=1\dots M}, \theta^m \sim p(\theta), X^m \sim p(X|\theta)$. This is the loss used in most of the SBI literature (Cranmer et al., 2020), as those are based around the possibility to be *likelihood-free*, and have a target density $p$ only implicitly defined by a simulator:

$$
\begin{aligned}
\Psi^\star &= \arg\min_{\Psi} \mathbb{E}_{X\sim p(X)}[\mathrm{KL}(p(\theta|X)||q_\Psi(\theta|X)] \\
&= \arg\min_{\Psi} \mathbb{E}_{X\sim p(X)}[\mathbb{E}_{\theta\sim p(\theta|X)}[\log p(\theta|X) - \log q_\Psi(\theta|X)]] \\
&= \arg\min_{\Psi} \mathbb{E}_{X\sim p(X)}[\mathbb{E}_{\theta\sim p(\theta|X)}[-\log q_\Psi(\theta|X)]] \\
&= \arg\min_{\Psi} \int p(X)\Big[\int -p(\theta|X)\log q_\Psi(\theta|X)d\theta\Big]dX \\
&= \arg\min_{\Psi} \int\int -p(X,\theta)\log q_\Psi(\theta|X)d\theta dX \\
&\approx \arg\min_{\Psi} \frac{1}{M}\times\sum_{m=1}^{M} -\log q_\Psi(\theta^m|X^m) \\
&\text{where } \theta^m \sim p(\theta), X^m \sim p(X|\theta)
\end{aligned}
\tag{C.6}
$$

- using a *reverse* KL divergence, benefiting from the access to a target joint density $p(X,\theta)$. The reverse KL loss is an amortized version of the classical ELBO expression (Blei et al., 2017). For training, one only needs to have access to a dataset $\{X^m\}_{m=1\dots M}, X^m \sim p(X)$ of points drawn from the generative HBM of interest. Indeed, the $\theta^m$ points are sampled from the variational distribution:

$$
\begin{aligned}
\Psi^\star &= \arg\min_{\Psi} \mathbb{E}_{X\sim p(X)}[\mathrm{KL}(q_\Psi(\theta|X)||p(\theta|X)] \\
&= \arg\min_{\Psi} \mathbb{E}_{X\sim p(X)}[\mathbb{E}_{\theta\sim q_\Psi(\theta|X)}[\log q_\Psi(\theta|X) - \log p(\theta|X)]] \\
&= \arg\min_{\Psi} \mathbb{E}_{X\sim p(X)}[\mathbb{E}_{\theta\sim q_\Psi(\theta|X)}[\log q_\Psi(\theta|X) - \log p(X,\theta) + \log p(X)]] \\
&= \arg\min_{\Psi} \mathbb{E}_{X\sim p(X)}[\mathbb{E}_{\theta\sim q_\Psi(\theta|X)}[\log q_\Psi(\theta|X) - \log p(X,\theta)]] \\
&= \arg\min_{\Psi} \int p(X)\Big[\int q_\Psi(\theta|X)[\log q_\Psi(\theta|X) - \log p(X,\theta)]d\theta\Big]dX \\
&= \arg\min_{\Psi} \int\int p(X)q_\Psi(\theta|X)[\log q_\Psi(\theta|X) - \log p(X,\theta)]d\theta dX \\
&\approx \arg\min_{\Psi} \frac{1}{M}\times\sum_{m=1}^{M} \log q_\Psi(\theta^m|X^m) - \log p(X^m,\theta^m) \\
&\text{where } X^m \sim p(X), \theta^m \sim q_\Psi(\theta|X)
\end{aligned}
\tag{C.7}
$$

As it more uniquely fits our setup and provided better results experimentally, we chose to focus on the usage of the *reverse* KL divergence. During our experiments, we also tested the usage of the *unregularized ELBO* loss:

$$
\Psi^\star = \arg\min_{\Psi} \frac{1}{M}\times\sum_{m=1}^{M} -\log p(X^m,\theta^m)
\tag{C.8}
$$
$$
\text{where } X^m \sim p(X), \theta^m \sim q_\Psi(\theta|X)
$$

|  | | Mean of analytical KL divergences from the theoretical posterior (low is good) | | | |
|  |  | Early stopping | | After convergence | |
| Loss | NaN runs | Mean | Std | Mean | Std |
|---|---|---|---|---|---|
| forward KL | 0 | 3847.7 | 5210.4 | 2855.3 | 4248.1 |
| unregularized ELBO | 0 | 6.6 | 0.7 | 6.2 | 0.9 |
| reverse KL | 2 | 12.3 | 19.8 | 3.0 | 4.1 |

Table 3: Convergence of the variational posterior to the analytical posterior over an early stopped training (200 batches) and after convergence (1000 batches) for the Gaussian random effects example

This formula differs from the one of the reverse KL loss by the absence of the term $q_\Psi(\theta^m|X^m)$, and is a converse formula to the one of the forward KL (in the sense that it permutes the roles of $q$ and $p$).

Intuitively, it posits our architecture as a pure sampling distribution that aims at producing points $\theta^m$ in regions of high joint density $p$. In that sense, it acts as a first moment approximation for the target posterior distribution (akin to MAP parameter regression). Experimentally, the usage of the *unregularized ELBO* loss provided fast convergence to a mode of the posterior distribution, with very low variance for the variational approximation.

We argue the possibility to use the *unregularized ELBO* loss as a *warm-up* before switching to the reverse KL loss, with the latter considered here as a regularization of the former. We introduce this training strategy as an example of the modularity of our approach, where one could transfer the rapid learning from one task (amortized mode finding) to another task (amortized posterior estimation).

**Graphical comparison**   In Figure C.1 we analyse the influence of these 3 different losses on the training of our posterior distribution, compared to the analytical ground truth. This example is typical of the relative behaviors induced on the variational distributions by each loss:

- The forward KL provides very erratic training, and results after several dozen epochs (several minutes) with a careful early stopping in posteriors with too large variance.

- The unregularized ELBO loss converges in less then 3 epochs (a couple dozen seconds), and provides posteriors with very low variance, concentrated on the MAP estimates of their respective parameters.

- The reverse KL converges in less 10 epochs (less than 3 minutes) and provides relevant variance.

**Losses convergence speed**   We analyse the relative convergence speed of our variational posterior to the analytical one (derived in eq. (C.5a)) when using the 3 aforementioned losses for training. To measure the convergence, we compute analytically the KL divergence between the variational posterior and the analytical one (every distribution being a Gaussian), summed for every distribution, and averaged over a validation dataset of size 2000.

We use a training dataset of size 2000, and for each loss repeated the training 20 times (batch size 10, 10 $\theta^m$ samples per $X^m$) for 10 epochs, resulting in 200 optimizer calls. This voluntary low number allows us to asses how close is the variational posterior to the analytical posterior after only a brief training. Results are visible in appendix C.4, showing a faster convergence for the *unregularized ELBO*. After 800 more optimizer calls, the tendency gets inverted and the *reverse KL* loss appears as the superior loss (though we still notice a larger variance).

The large variance in the results may point towards the need for adapted training strategies involving Learning rate decay and/or scheduling (Kucukelbir et al., 2016), an extension that we leave for future work.

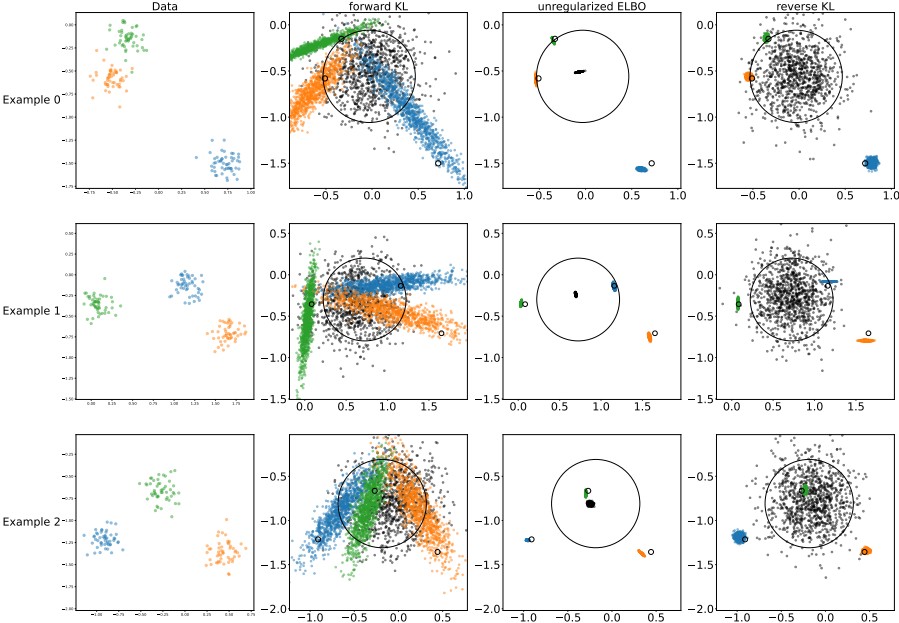

Figure C.1: Graphical results on the Gaussian random effects example for our architecture trained using 3 different losses. Rows represent 3 different data points. Left column represents data, with colors representing 3 different groups. Other columns represent posterior samples for $\mu$ (black) and $\mu^1, \mu^2, \mu^3$. Visually, posterior samples $\mu^1, \mu^2, \mu^3$ should be concentrated around the mean of the data points with the same color, and the black points $\mu$ should be repartitioned around the mean of the 3 group means (with a shift towards 0 due to the prior). Associated with the posterior samples are analytical solutions (thin black circles), centered on the analytical MAP point, and whose radius correspond to 2 times the standard deviation of the analytical posterior: 95 % of the draws from a posterior should fall within the corresponding circle.

In section 2.3, for the reverse KL loss, we approximate expectations using Monte Carlo integration.
We further train our architecture using minibatch gradient descent, as opposed to stochastic gradient
descent as proposed by Kucukelbir et al. (2016). An interesting hyper-parametrization of our system
resides in the effective batch size of our training, that depends upon:

- the size of the mini batches, determining the number of $X^m$ points considered in parallel

- the number of $\theta^m$ draws per $X^m$ point, that we use to approximate the gradient in the
  ELBO

More formally, we define a computational budget as the relative allocation of a constant effective
batch size $\text{batch size} \times \theta$ draws per X between $\text{batch size}$ and $\theta$ draws per X.

To analyse the effect of the computational budget on training, we use a dataset of size 1000, and run
experiment 20 times over the same number of optimizer calls with the same effective batch size per
call. Results can be seen in Fig. C.2. From this experiment we can draw the following conclusions:

- we didn't witness massive difference in the global convergence speed across computational
  budgets

- the bigger the budget we allocate to the sampling of multiple $\theta^m$ per point $X^m$ (effectively
  going towards a stochastic training in terms of the points $X^m$), the more erratic is the loss
  evolution

- the bigger the budget we allocate to the $X^m$ batch size, the more stable is the loss evolution,
  but our interpretation is that the resulting reduced number of $\theta^m$ draws per $X^m$ augments
  the risk of an instability resulting in a NaN run

Experimentally, we obtained the best results by evenly allocating our budget to the $X^m$ batch size
and the number of $\theta^m$ draws per $X^m$ point (typically, 32 and 32 respectively for an effective batch
size of 1024). Overall, in the amortized setup, our experiment stand as a counterpoint to those of
Kucukelbir et al. (2016) who pointed towards the case of a single $\theta^m$ draw per point $X^m$ as their
preferred hyper-parametrization.

# D    COMPLEMENTS TO THE GAUSSIAN MIXTURE WITH RANDOM EFFECTS
EXPERIMENT: FURTHER POSTERIOR ANALYSIS

This section is a complement to the experiment described in section 3.4, we thus consider the model
described in eq. (8a). We explore the complexity of the theoretical posterior for this experiment.

## D.1    DESCRIPTORS, INPUTS TO ADAVI

We can analyse the model described in eq. (8a) using the descriptors defined in eq. (A.1). Those
descriptors constitute the inputs our methodology needs to automatically derive the *dual* architecture

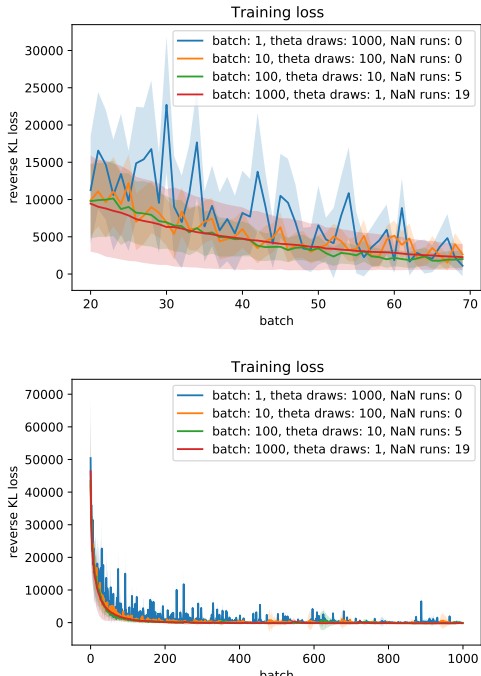

Figure C.2: Loss evolution across batches for different computational budgets. All experiments are designed so that to have the same number of optimizer calls (meaning that $\text{batch size} \times \text{epochs} = 1000$) and the same effective batch size (meaning that $\text{batch size} \times \theta \text{ draws per X} = 1000$). Every experiment is run 20 times, error bands showing the standard deviation of the loss at the given time point. Note that the blue line (batch size 1, 1000 $\theta$ draws per $X$) is more erratic than the other ones (even after a large number of batches). On the other hand, the red line (batch size 1000, 1 $\theta$ draws per $X$) is more stable, but 19 out of 20 runs ultimately resulted in an instability

from the generative HBM:

$$\mathcal{V} = \{M^L, M^{L,G}, \Pi^G, X\}$$
$$\mathsf{P} = \{\mathcal{P}_0, \mathcal{P}_1\}$$
$$\mathrm{Card} = \{\mathcal{P}_0 \mapsto N, \mathcal{P}_1 \mapsto G\}$$
$$\mathrm{Hier} = \{M^L \mapsto 2, M^{L,G} \mapsto 1, \Pi^G \mapsto 1, X \mapsto 0\}$$
$$\mathrm{Shape} = \{M^L \mapsto (L, D), M^{L,G} \mapsto (L, D), \Pi^G \mapsto (L,), X \mapsto (D,)\}$$
$$\mathrm{Link} = \{ \tag{D.9}$$
$$M^L \mapsto \mathrm{Reshape}((LD,) \to (L, D)),$$
$$M^{L,G} \mapsto \mathrm{Reshape}((LD,) \to (L, D)),$$
$$\Pi^G \mapsto \mathrm{SoftmaxCentered}((L-1,) \to (L,)),$$
$$X \mapsto \mathrm{Identity}$$
$$\}$$

For the definition of the SoftmaxCentered link function, see Dillon et al. (2017).

## D.2 THEORETICAL POSTERIOR RECOVERY IN THE GAUSSIAN MIXTURE RANDOM EFFECTS MODEL

We further analyse the complexity of model described in section 3.4.

Due to the label switching problem (Jasra et al., 2005), the relative position of the L mixture components in the D space is arbitrary. Consider a non-degenerate example like the one in Fig. D.3, where the data points are well separated in 3 blobs (likely corresponding to the $L = 3$ mixture components). Since there is no deterministic way to assign component $l = 1$ unequivocally to a blob of points, the marginalized posterior distribution for the position of the component $l = 1$ should be multi-modal, with -roughly- a mode placed at each one of the 3 blobs of points. This posterior would be the same for the components $l = 2$ and $l = 3$. In truth, the posterior is even more complex than this 3-mode simplification, especially when the mixture components are closer to each other in 2D (and the grouping of points into draws from a common component is less evident).

In Fig. D.3, we note that our technique doesn't recover this multi-modality in its posterior, and instead assigns different posterior components to different blobs of points. Indeed, when plotting only the posterior samples for the first recovered component $l = 1$, all points are concentrated around the bottom-most blob, and not around each blob like the theoretical posterior would entail (see Fig. D.3 second row).

This behavior most likely represents a local minimum in the reverse KL loss that is common to many inference techniques (for instance consider multiple non-mixing chains for MCMC in Jasra et al., 2005). We note that training in forward KL wouldn't provide such a flexibility in that setup, as it would enforce the multi-modality of the posterior, even at the cost of an overall worst result (as it is the case for NPE-C and SNPE-C in table 1. Indeed, let's imagine that our training dataset features $M'$ draws similar to the one in Fig. D.3. Out of randomness, the labelling $l$ of the 3 blobs of points would be permuted across those $M'$ examples. A forward-KL-trained density estimator would then most likely attempt to model a multi-modal posterior.

Though it is not similar to the theoretical result, we argue that our result is of experimental value, and close to the intuition one forms of the problem: using our results one can readily estimate the original components for the mixture. Indeed, for argument's sake, say we would recover the theoretical, roughly trimodal posterior. To recover the original mixture components, one would need to split the 3 modes of the posterior and arbitrarily assign a label $l$ to each one of the modes. In that sense, our posterior naturally features this splitting, and can be used directly to estimate the $L = 3$ mixture components.

## E COMPLEMENTS TO THE NEUROIMAGING EXPERIMENT

This section is a complement to the experiment described in section 3.5, we thus consider the model described in eq. (E.10a) and eq. (E.10b). We present a toy dimension version of our experiment, use-

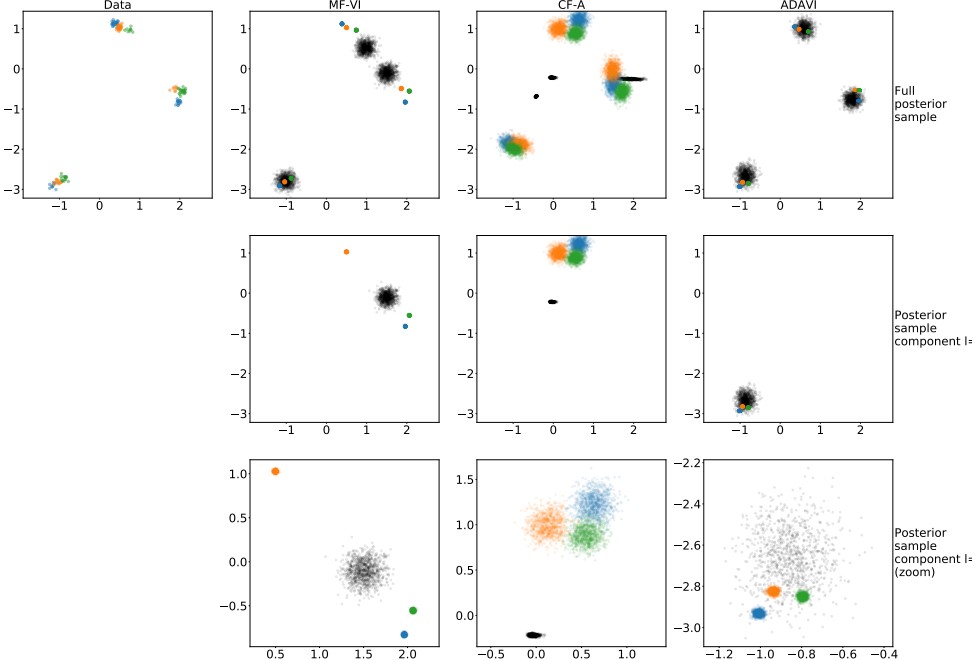

Figure D.3: Graphical comparison for various methods on the Gaussian mixture with random effects example. First column represents a non-degenerate data point, with colored points corresponding to $[x^{1,1}, ..., x^{1,N}]$, $[x^{2,1}, ..., x^{2,N}]$, $[x^{3,1}, ..., x^{3,N}]$. Note the distribution of the points around in 3 multi-colored groups (population components), and 3 colored sub-groups per group (group components). All other columns represent the posterior samples for population mixture components $\mu_1, \ldots, \mu_3$ (black) and group mixture components $\mu_1^1, \ldots, \mu_3^1, \mu_1^2, \ldots, \mu_3^2, \mu_1^3, \ldots, \mu_3^3$. Second column represents the results of a non-amortized MF-VI (best ELBO score across all random seeds): results are typical of a local minimum for the loss. Third column represents the result of an amortized CF-A (best amortized ELBO). Last column represents our amortized ADAVI technique (best amortized ELBO). First row represents the full posterior samples. Second and third row only represents the first mixture component samples (third row zooms in on the data). Notice how neither technique recovers the actual multi-modality of the theoretical posterior. We obtain results of good experimental value, usable to estimate likely population mixture components.

ful to build an intuition of the problem. We also present implementation details for our experiment, and additional neuroimaging results.

## E.1 Neuroimaging context

The main goal of Kong et al. (2019) is to address the classical problem in neuroscience of estimating population commonalities along with individual characteristics. In our experiment, we are interested in parcelling the region of left inferior frontal gyrus (IFG). Anatomically, the IFG is decomposed in 2 parts: pars opercularis and triangularis. Our aim is to reproduce this binary split from a functional connectivity point of view, an open problem in neuroscience (see e.g. Heim et al., 2009).

As Kong et al. (2019), we consider a population of S=30 subjects, each with $T = 4$ acquisition sessions, from the Human Connectome Project dataset (Van Essen et al., 2012). The fMRI connectivity between a cortical point and the rest of the brain, split in $D = 1,483$ regions, is represented as a vector of length D with each component quantifying the temporal correlation of blood-oxygenation between the point and a region. A main hypothesis of Kong et al. (2019), and the fMRI field, is that the fMRI connectivity of points belonging to the same parcel share a similar connectivity pattern or correlation vector. Following Kong et al. (2019), we represent D-dimensional correlation vectors as RVs on the positive quadrant of the $D$-dimensional unit-sphere. We do this efficiently assuming they have a $\mathcal{L}$-normal distribution, or Gaussian under the transformation of the link function $\mathcal{L}(x) = \sqrt{\text{SoftmaxCentered}(x)}$ (Dillon et al., 2017):

$$
\begin{aligned}
\pi,\ s^-,\ s^+ &= 2,\ -10,\ 8 \\
\mathcal{L}^{-1}(\mu_l^g) &\sim \mathcal{N}(\vec{0}_{D-1}, \Sigma_g) & M^L &= [\mu_l^g]^{l=1...L} \\
\log(\epsilon_l) &\sim \mathcal{U}(s^-, s^+) & E^L &= [\epsilon_l]^{l=1...L} \\
\mathcal{L}^{-1}(\mu_l^s) \mid \mu_l^g,\ \epsilon_l &\sim \mathcal{N}(\mathcal{L}^{-1}(\mu_l^g), \epsilon_l^2) & M^{L,S} &= [\mu_l^s]^{l=1...L}_{s=1...S} \\
\log(\sigma_l) &\sim \mathcal{U}(s^-, s^+) & \Sigma^L &= [\sigma_l]^{l=1...L} \\
\mathcal{L}^{-1}(\mu_l^{s,t}) \mid \mu_l^s,\ \sigma_l &\sim \mathcal{N}(\mathcal{L}^{-1}(\mu_l^s), \sigma_l^2) & M^{L,S,T} &= [\mu_l^{s,t}]^{l=1...L}_{\substack{s=1...S \\ t=1...T}} \\
\log(\kappa) &\sim \mathcal{U}(s^-, s^+) \\
\Pi &\sim \text{Dir}([\pi] \times L)
\end{aligned}
\tag{E.10a}
$$

$$
\mathcal{L}^{-1}(X_n^{s,t}) \mid [\mu_1^{s,t}, \ldots, \mu_L^{s,t}], \kappa, \Pi \sim \text{Mix}(\Pi, [N(\mathcal{L}^{-1}(\mu_1^{s,t}), \kappa^2), \ldots, N(\mathcal{L}^{-1}(\mu_L^{s,t}), \kappa^2)])
$$
$$
X = [X_n^{s,t}]^{s=1...S}_{\substack{t=1...T \\ n=1...N}}
$$
$$
\tag{E.10b}
$$

Our aim is therefore to identify $L = 2$ functional networks that would produce a functional parcellation of the studied IFG section. In this setting, the parameters $\theta$ of interest are the networks $\mu$. Instead of the complex EM computation derived in Kong et al. (2019), we perform full-posterior inference for those parameters using our automatically derived architecture.

## E.2 Descriptors, inputs to ADAVI

We can analyse the model described in eq. (E.10a) and eq. (E.10b) using the descriptors defined in eq. (A.1). Those descriptors constitute the inputs our methodology needs to automatically derive the

*dual* architecture from the generative HBM:

$$
\begin{aligned}
\mathcal{V} &= \{M^L, E^L, M^{L,S}, \Sigma^L, M^{L,S,T}, \kappa, \Pi, X\} \\
\texttt{P} &= \{\mathcal{P}_0, \mathcal{P}_1, \mathcal{P}_2\} \\
\mathrm{Card} &= \{\mathcal{P}_0 \mapsto N, \mathcal{P}_1 \mapsto T, \mathcal{P}_2 \mapsto S\} \\
\mathrm{Hier} &= \{M^L \mapsto 3, E^L \mapsto 3, M^{L,S} \mapsto 2, \Sigma^L \mapsto 3, M^{L,S,T} \mapsto 1, \kappa \mapsto 3, \Pi \mapsto 3, X \mapsto 0\} \\
\mathrm{Shape} &= \{ \\
M^L &\mapsto (L, D), \\
E^L &\mapsto (L, ), \\
M^{L,S} &\mapsto (L, D), \\
\Sigma^L &\mapsto (L, ), \\
M^{L,S,T} &\mapsto (L, D), \\
\kappa &\mapsto (1, ), \\
\Pi &\mapsto (L, ), \\
X &\mapsto (D, ) \\
&\} \\
\mathrm{Link} &= \{ \\
M^L &\mapsto \mathcal{L} \circ \mathrm{Reshape}((LD, ) \to (L, D)), \\
E^L &\mapsto \mathrm{Exp}, \\
M^{L,S} &\mapsto \mathcal{L} \circ \mathrm{Reshape}((LD, ) \to (L, D)), \\
\Sigma^L &\mapsto \mathrm{Exp}, \\
M^{L,S,T} &\mapsto \mathcal{L} \circ \mathrm{Reshape}((LD, ) \to (L, D)), \\
\kappa &\mapsto \mathrm{Exp}, \\
\Pi &\mapsto \mathrm{SoftmaxCentered}((L-1, ) \to (L, )), \\
X &\mapsto \mathcal{L} \\
&\}
\end{aligned}
\tag{E.11}
$$

### E.3    EXPERIMENT ON MS-HBM MODEL ON TOY DIMENSIONS

To get an intuition of the behavior of our architecture on the MS-HBM model, we consider the following toy dimensions reproduction of the model:

$$
\begin{aligned}
N, T, S, D, L &= 50, 2, 2, 2, 2 \\
g^-, g^+ &= -4, 4 \\
\kappa^-,\, \kappa^+,\, \sigma^-,\, \sigma^+,\, \epsilon^-,\, \epsilon^+ &= -4,\, -4,\, -3,\, -3,\, -2,\, -2,\, -1 \\
\pi &= 2 \\
\mathcal{L}^{-1}(\mu_l^g) &\sim \mathcal{U}(-g^-, g^+) \\
\log(\epsilon_l) &\sim \mathcal{U}(\epsilon^-, \epsilon^+) \\
\mathcal{L}^{-1}(\mu_l^s)|\mu_l^g,\, \epsilon_l &\sim \mathcal{N}(\mathcal{L}^{-1}(\mu_l^g), \epsilon_l^2) \\
\log(\sigma_l) &\sim \mathcal{U}(\sigma^-, \sigma^+) \\
\mathcal{L}^{-1}(\mu_l^{s,t})|\mu_l^s,\, \sigma_l &\sim \mathcal{N}(\mathcal{L}^{-1}(\mu_l^s), \sigma_l^2) \\
\log(\kappa) &\sim \mathcal{U}(\kappa^-, \kappa^+) \\
\Pi &\sim \mathrm{Dir}([\pi] \times L) \\
\mathcal{L}^{-1}(X_n^{s,t})|[\mu_1^{s,t}, \ldots, \mu_L^{s,t}], \kappa, \Pi &\sim \mathrm{Mix}(\Pi, [\mathcal{N}(\mathcal{L}^{-1}(\mu_1^{s,t}), \kappa^2), \ldots, \mathcal{L}(\mathcal{L}^{-1}(\mu_L^{s,t}), \kappa^2)])
\end{aligned}
\tag{E.12}
$$

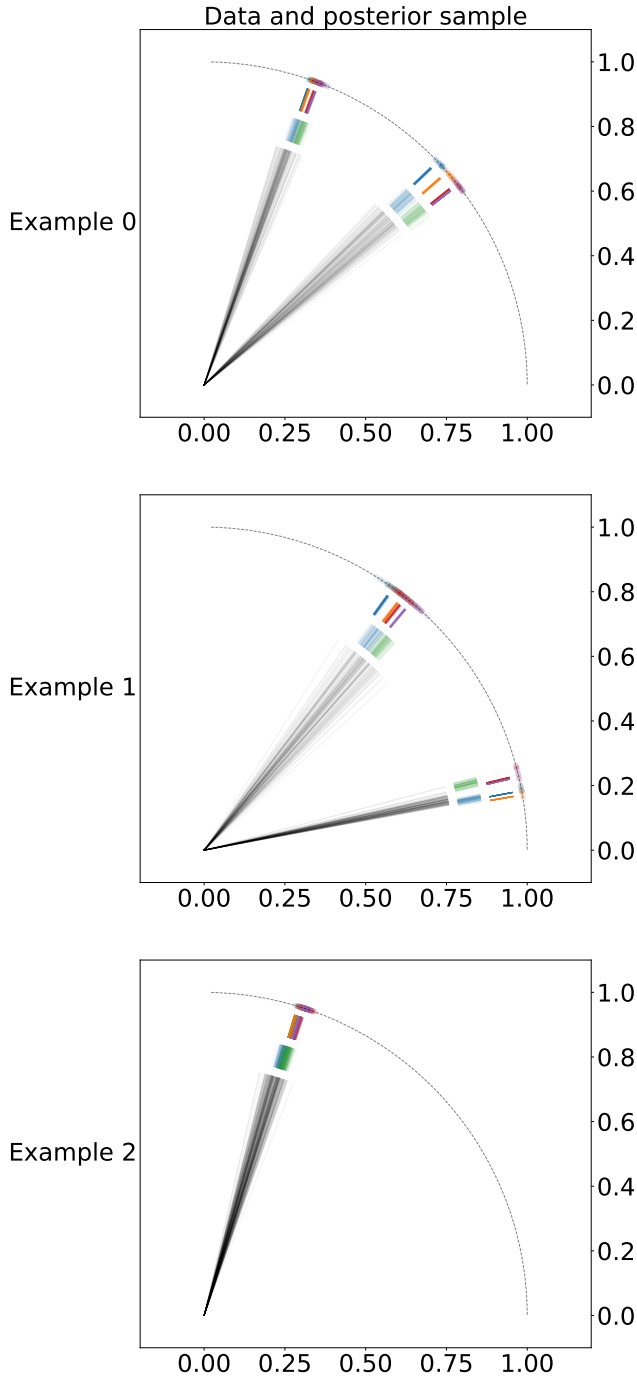

Figure E.4: Visual representation of our results on a synthetic MS-HBM example. Data is represented as colored points on the unit positive quadrant, each color corresponding to a subject $\times$ session. Samples from posterior distributions are represented as concentric colored markings. Just below the data points are $\mu^{s,t}$ samples. Then samples of $\mu^s$. Then samples of $\mu^g$ (black lines). Notice how the $\mu$ posteriors are distributed around the angle bisector of the arc covered by the points at the subsequent plate.

The results can be visualized on Fig. E.4. This experiment shows the expressivity we gain from the usage of link functions.

**Main implementation differences with the original MS-HBM model**  Our implementation of the MS-HBM (eq. (E.10a) and eq. (E.10b)) contains several notable differences with the original one from Kong et al. (2019):

- we model $\mu$ distributions as Gaussians linked to the positive quadrant of the unit sphere via the function $\mathcal{L}$. In the orignal model, RVs are modelled using *Von Mises Fisher* distributions. Our choice allows us to express the entirety of the connectivity vectors (that only lie on a portion of the unit sphere). However, we also acknowledge that the densities incurred by the 2 distributions on the positive quadrant of the unit sphere are different.

- we forgo any spatial regularization, and also the assumption that the parcellation of a given subject $s$ should be constant across sessions $t$. This is to streamline our implementation. Adding components to the loss optimized at training could inject those constraints back into the model, but this was not the subject of our experiment, so we left those for future work.

**Data pre-processing and dimensionality reduction**  Our model was able to run on the full dimensionality of the connectivity, $D^0 = 1483$. However, we obtained better results experimentally when further pre-processing the used data down to the dimension $D^1 = 141$. The displayed results in Fig. 4 are the ones resulting from this dimensionality reduction:

1. we projected the $(S, T, N, D^0)$ $X$ connectome (lying on the $D^0$ unit sphere) to the unbounded $\mathbb{R}^{D^0-1}$ space using the function $\mathcal{L}$

2. in this $\mathbb{R}^{D^0-1}$ space, we performed a Principal Component Analysis (PCA) to bring us down to $D^1 - 1 = 140$ dimensions responsible for 80% of the explained data variance

3. in the resulting $\mathbb{R}^{D^1-1}$ space, we calculated the mean of all the connectivity points, and their standard deviation, and used both to whiten the data

4. from the whitened data, we calculated the Ledoit-Wolf regularised covariance (Ledoit & Wolf, 2004), that we used to construct the $\Sigma_g$ matrix used in eq. (E.10a)

5. finally, we projected the whitened data onto the unit sphere in $D^1 = 141$ dimensions via the function $\mathcal{L}$

To project our results back to the original $D^0$ space, we simply ran back all the aforementioned steps. Our prior for $\mu^g$ has been carefully designed so has to sample connectivity points in the vicinity of the data point of interest. Our implementation is therefore in spirit close to SBI (Cranmer et al., 2020; Papamakarios et al., 2019b; Greenberg et al., 2019; Thomas et al., 2020) that aims at obtaining an amortized posterior only in the relevant data regime.

**Mutli-step training strategy**  In appendix F.2 we describe our conditional density estimators as the stacking of a MAF (Papamakarios et al., 2018) on top of a diagonal-scale affine block. To accelerate the training of our architecture and minimize numerical instability (resulting in NaN evaluations of the loss) we used the following 3-step training strategy:

1. we only trained the *shift* part of our affine block into a Maximum A Posteriori regression setup. This can be viewed as the amortized fitting of the first moment of our posterior distribution

2. we trained both the *shift* and *scale* of our affine block using an *unregularized ELBO* loss. This is to rapidly bring the variance of our posterior to relevant values

3. we then trained our full posterior (*shift* and *scale* of our affine block, in addition to our MAF block) using the *reverse KL* loss.

This training strategy shows the modularity of our approach and the transfer learning capabilities already introduced in appendix C.4. Loss evolution can be seen in Fig. E.5

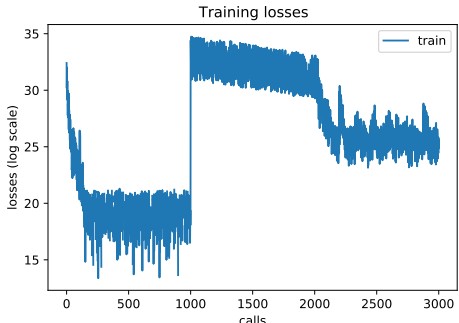

Figure E.5: 3-step loss evolution across epochs for the MS-HBM ADAVI training. Losses switch are visible at epochs 1000 and 2000. Training was run for a longer period after epoch 3000, with no significant results difference.

**Soft labelling**  In eq. (E.10a) and eq. (E.10b) we define $\mu$ variables as following Gaussian distributions in the latent space $\mathbb{R}^{D^1-1}$. This means that, considering a vertex $X_n^{s,t}$ and a session network $\mu_k^{s,t}$, the squared Euclidean distance between $\mathcal{L}^{-1}(X_n^{s,t})$ and $\mathcal{L}^{-1}(\mu_k^{s,t})$ in the latent space $\mathbb{R}^{D^1-1}$ is proportional to the log-likelihood of the point $\mathcal{L}^{-1}(X_n^{s,t})$ for the mixture component $k$:

$$\|\mathcal{L}^{-1}(X_n^{s,t}) - \mathcal{L}^{-1}(\mu_k^{s,t})\|^2 = \log p(X_n^{s,t}|l = k) + C(\kappa) \tag{E.13}$$

Note that $\kappa$ is the same for both networks. Additionally, considering Bayes theorem:

$$\log p(l = k|X_n^{s,t}) = \log p(X_n^{s,t}|l = k) + \log p(l = k) - \log p(X_n^{s,t})$$

$$\log \frac{p(l = 0|X_n^{s,t})}{p(l = 1|X_n^{s,t})} = \log p(X_n^{s,t}|l = 0) - \log p(X_n^{s,t}|l = 1) + \log \frac{p(l = 0)}{p(l = 1)} \tag{E.14}$$

Where $\log p(X_n^{s,t}|l = k)$ can be obtained through eq. (E.13) and $\log p(l = k)$ via draws from the posterior of $\Pi$ (see eq. (E.10a)). To integrate those equations, we used a Monte Carlo procedure.

### E.5  ADDITIONAL NEUROIMAGING RESULTS

#### E.5.1  SUBJECT-LEVEL PARCELLATION

As pointed out in section 3.5, the MS-HBM model aims at representing the functional connectivity of the brain at different levels, allowing for estimates of population characteristics and individual variability (Kong et al., 2019). It is of experimental value to compare the parcellation for a given subject, that is to say the soft label we give to a vertex $X_n^{s,t}$, and how this subject parcellation can deviate from the population parcellation. Those differences underline how an individual brain can have a unique local organization. Similarly, we can obtain the subject networks $\mu^s$ and observe how those can deviate from the population networks $\mu^g$. Those results underline how a given subject can have his own connectivity, or, very roughly, his own "wiring" between different areas of the brain. Results can be seen in Fig. E.6.

#### E.5.2  COMPARISON OF LABELLING WITH THE MF-VI RESULTS

We can compare the subject-level parcellation resulting from the latent networks recovered using the ADAVI vs the MF-VI method. The result, for the same subjects as the previous section, can be seen in Fig. E.7, where the difference in ELBO presented in our main text translates into marginal differences for our downstream task of interest.

## F  BASELINES FOR EXPERIMENTS

### F.1  BASELINE CHOICE

In this section we justify further the choice of architectures presented as baselines in the our experiments:

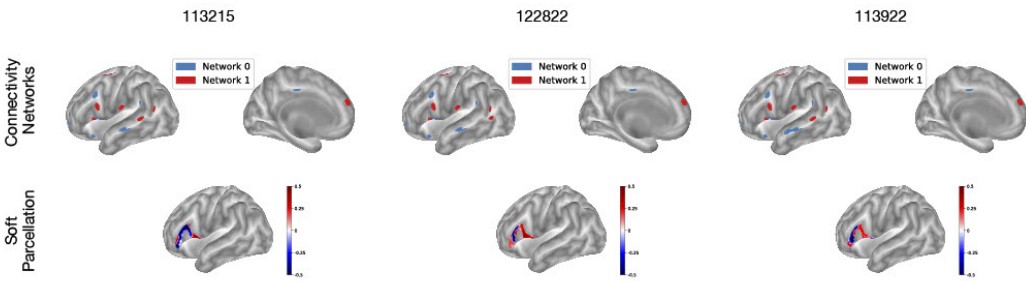

Figure E.6: Subject-level parcellations and networks. For 3 different HCP subjects, we display the individual parcellation (on the bottom) and the individual $\mu^s$ networks (on the top). Note how the individual parcellations, though showing the same general split between the *pars opercularis* and *pars triangularis*, slightly differ from each other and from the population parcellation (Fig. 4). Similarly, networks $\mu^s$ differ from each other and from the population networks $\mu^g$ (Fig. 4) but keep their general association to semantic/phonologic processing (0, in blue) and language production (1, in red) (Heim et al., 2009; Zhang et al., 2020). To be able to model and display such variability is one of the interests of models like the MS-HBM (Kong et al., 2019).

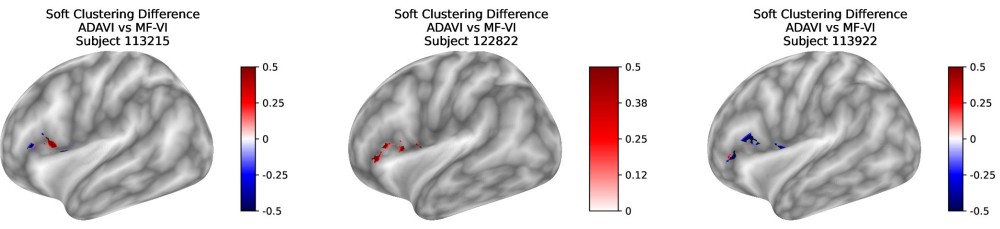

Figure E.7: Gap in labelling between ADAVI and MF-VI results. Following eq. (E.14), we compute the difference in latent space between the odds for the ADAVI and MF-VI techniques, before applying a sigmoid function. Differences are marginal, and interestingly located at the edges between networks, where the labelling is less certain.

- *Mean Field VI* (MF-VI) (Blei et al., 2017). This methods stands as a common-practice non-amortized baseline, is fast to compute, and due to our choice of conjugate examples (see section 3.2) can be considered as a proxy to the ground truth posterior. We implemented MF-VI in its usual setup, fitting to the posterior a distribution of the prior's parametric form;

- *(Sequential) Neural Posterior Estimation* (SNPE-C) architecture (Greenberg et al., 2019). NPE-C is an typical example from the SBI literature (Cranmer et al., 2020), and functions as a *likelihood-free*, *black box* method. Indeed, NPE-C is trained using forward KL (samples from the latent parameters), and is not made "aware" of any structure in the problem. NPE-C fits a single normalizing flow over the entirety of the latent parameter space, and its number of weights scales quadratically with the parameter space size. When ran over several simulation rounds, the method becomes *sequential* (SNPE-C), specializing for a certain parameter regime to improve performance, but loosing amortization in the process;

- *Total Latent Space Flow* (TLSF) architecture (Rezende & Mohamed, 2016). Following the original normalizing flow implementation from Rezende & Mohamed (2016), we posit TLSF as a counterpoint to SNPE-C. Like SNPE-C, TLSF fits a single normalizing flow over the entirety of the latent parameter space, and is not made "aware" of the structure of the model. But contrary to SNPE-C, TLSF is trained using reverse KL and benefits from the presence of a likelihood function. We can use TLSF in a non-amortized setup (TLSF-NA), or in an amortized setup (TLSF-A) trough an observed data encoder conditioning the single normalizing flow;

- *Cascading Flows* (CF) (Ambrogioni et al., 2021b). CF is an example of a structure-aware, prior-aware VI method, trained using reverse KL. By design, its number of weights scales linearly with the plate's cardinalities. CF can be ran both in a non-amortized (CF-NA) and amortized (CF-A) setup, with the introduction of amortization through observed data encoders in the auxiliary graph. As a structure-aware amortized architecture, CF-A is our main point of comparison in this section;

## F.2 IMPLEMENTATION DETAILS

In this section, we describe with precision and per experiment the implementation details for the architectures described in appendix F.1. We implemented algorithms in Python, using the Tensorflow probability (TFP, Dillon et al., 2017) and Simulation Based Inference (SBI, Tejero-Cantero et al., 2020) libraries. For all experiments in TFP, we used the Adam optimizer (Kingma & Ba, 2015). For normalizing flows, we leveraged Masked Autoregressive Flow (MAF, Papamakarios & Murray, 2018).

For all experiments:

- *Mean Field VI* (MF-VI) (Blei et al., 2017). We implemented MF-VI in TFP. The precise form of the variational family is described below for each experiment.

- *Sequential Neural Posterior Estimation* (SNPE-C) architecture (Greenberg et al., 2019). We implemented SNPE-C with the SBI library, using the default parameters proposed by the API. Simulations were ran over 5 rounds, to ensure maximal performance. We acknowledge that this choice probably results in an overestimate of the runtime for the algorithm. To condition the density estimation based on the observed data, we designed an encoder that is a variation of our Hierarchical Encoder (see section 2.2). Its architecture is the same as HE -the hierarchical stacking of 2 Set Transformers (Lee et al., 2019)- but the encoder's output is the concatenation of the G per-group encodings with the population encodings. This encoder is therefore parsimoniously parametrized, and adapted to the structure of the problem.

- *Neural Posterior Estimation* (NPE-C). Though we acknowledge that NPE-C can be implemented easily using the SBI library, we preferred to use our own implementation, notably to have more control over the runtime of the algorithm. We implemented the algorithm using TFP. We used the same encoder architecture as for SNPE-C.

- *Total Latent Space Flow* (TLSF). We implemented TLSF using TFP. Our API is actually the same as for NPE-C, since TLSF-A and NPE-C only differ by their training loss.

- *Cascading Flows* (CF) (Ambrogioni et al., 2021b). We implemented our own version of Cascading Flows, using TFP, and having consulted with the authors. An important implementation detail that is not specified explicitly in Ambrogioni et al. (2021b) (whose notations we follow here) is the implementation of the target distribution over the auxiliary variables $r$, notably in the amortized setup. Following the authors specifications during our discussion, we implemented $r$ as the Mean Field distribution $r = \prod_j p_j(\epsilon_j)$.

- ADAVI (ours). We implemented ADAVI using TFP.

Regarding the training data:

- All amortized methods were trained over a dataset of $20,000$ samples

- All non-amortized methods except SNPE-C were trained on a single data point (separately for 20 different data points)

- SNPE-C was trained over 5 rounds of simulations, with $1000$ samples per round, for an effective dataset size of $5000$

For the non conjugate experiment (see section 3.2):

- MF-VI: variational distribution is

$$q = Gamma(a; \text{concentration=}V_{(D,)}, \text{rate=} \text{Softplus}(V_{(1,)}))$$

  We used the Adam optimizer with a learning rate of $10^{-2}$. The optimization was ran for $20,000$ steps, with a sample size of 32.

- CF: auxiliary size 8, observed data encoders with 8 hidden units. Minibatch size 32, 32 theta draws per X point (see appendix C.5), Adam ($10^{-2}$), 40 epochs using a reverse KL loss.

- ADAVI: NF with 1 Affine block with triangular scale, followed by 1 MAF with $[32, 32, 32]$ units. HE with embedding size 8, 2 modules with 2 ISABs (2 heads, 8 inducing points), 1 PMA (seed size 1), 1 SAB and 1 linear unit each. Minibatch size 32, 32 theta draws per X point (see appendix C.5), Adam ($10^{-3}$), 40 epochs using a reverse KL loss.

For the Gaussian random effects experiment (see section 3.3):

- MF-VI: variational distribution is

$$q = \mathcal{N}(\mu; \text{mean=}V_{(D,)}, \text{std=} \text{Softplus}(V_{(1,)}))$$
$$\times \mathcal{N}([\mu^1, ..., \mu^G]; \text{mean=}V_{(G,D)}, \text{std=} \text{Softplus}(V_{(1,)}))$$

  We used the Adam optimizer with a learning rate of $10^{-2}$. The optimization was ran for $10,000$ steps, with a sample size of 32.

- SNPE-C: 5 MAF blocks with 50 units each. Encoder with embedding size 8, 2 modules with 2 SABs (4 heads) and 1 PMA (seed size 1) each, and 1 linear unit. See SBI for optimization details.

- NPE-C: 1 MAF with $[32, 32, 32]$ units. Encoder with embedding size 8, 2 modules with 2 ISABs (2 heads, 8 inducing points), 1 PMA (seed size 1), 1 SAB and 1 linear unit each. Minibatch size 32, 15 epochs with Adam ($10^{-3}$) using a forward KL loss.

- TLSF: same architecture as NPE-C. TLSF-A: minibatch size 32, 32 theta draws per X point (see appendix C.5), 15 epochs with Adam ($10^{-3}$) using a reverse KL loss. TLSF-NA: minibatch size 1, 32 theta draws per X point (see appendix C.5), 1500 epochs with Adam ($10^{-3}$) using a reverse KL loss.

- CF: auxiliary size 16, observed data encoders with 16 hidden units. CF-A: minibatch size 32, 32 theta draws per X point (see appendix C.5), Adam ($10^{-3}$), 40 epochs using a reverse KL loss. CF-NA: minibatch size 1, 32 theta draws per X point (see appendix C.5), Adam ($10^{-2}$), 1500 epochs using a reverse KL loss.

- ADAVI: NF with 1 Affine block with triangular scale, followed by 1 MAF with $[32, 32, 32]$ units. HE with embedding size 8, 2 modules with 2 ISABs (2 heads, 8 inducing points), 1 PMA (seed size 1), 1 SAB and 1 linear unit each. Minibatch size 32, 32 theta draws per X point (see appendix C.5), Adam ($10^{-3}$), 10 epochs using an unregularized ELBO loss, followed by 10 epochs using a reverse KL loss.

For the Gaussian mixture with random effects experiment (see section 3.4):

- MF-VI: variational distribution is

$$
\begin{aligned}
q = \ &\mathcal{N}([\mu_1, \ldots, \mu_L]; \text{mean=} V_{(L,D,)}, \text{std=} \text{Softplus}(V_{(1,)})) \\
&\times \mathcal{N}([[\mu_1^1, \ldots, \mu_L^1], ..., [\mu_1^G, ..., \mu_L^G]]; \text{mean=} V_{(G,L,D)}, \text{std=} \text{Softplus}(V_{(1,)})) \\
&\times \text{Dir}(\text{concentration=} \text{Softplus}(V_{(G,L)}))
\end{aligned}
$$

  We used the Adam optimizer with a learning rate of $10^{-2}$. The optimization was ran for $10,000$ steps, with a sample size of 32.

- SNPE-C: 5 MAF blocks with 50 units each. Encoder with embedding size 8, 2 modules with 2 SABs (4 heads) and 1 PMA (seed size 1) each, and 1 linear unit. See SBI for optimization details.

- NPE-C: 1 MAF with $[32, 32, 32]$ units. Encoder with embedding size 8, 2 modules with 2 ISABs (2 heads, 8 inducing points), 1 PMA (seed size 1), 1 SAB and 1 linear unit each. Minibatch size 32, 20 epochs with Adam ($10^{-3}$) using a forward KL loss.

- TLSF-A: same architecture as NPE-C, minibatch size 32, 32 theta draws per X point (see appendix C.5), 250 epochs with Adam ($10^{-3}$) using a reverse KL loss.

- TLSF-NA: same NF architecture as NPE-C. Encoder with embedding size 16, 2 modules with 2 ISABs (2 heads, 8 inducing points), 1 PMA (seed size 1), 1 SAB and 1 linear unit each. Minibatch size 1, 32 theta draws per X point (see appendix C.5), 1000 epochs with Adam ($10^{-3}$) using a reverse KL loss.

- CF: auxiliary size 8, observed data encoders with 8 hidden units. CF-A: minibatch size 32, 32 theta draws per X point (see appendix C.5), Adam ($10^{-3}$), 200 epochs using a reverse KL loss. CF-NA: minibatch size 1, 32 theta draws per X point (see appendix C.5), Adam ($10^{-2}$), 1500 epochs using a reverse KL loss.

- ADAVI: NF with 1 Affine block with diagonal scale, followed by 1 MAF with $[32]$ units. HE with embedding size 16, 2 modules with 2 ISABs (4 heads, 8 inducing points), 1 PMA (seed size 1), 1 SAB and 1 linear unit each. Minibatch size 32, 32 theta draws per X point (see appendix C.5), Adam ($10^{-3}$), 50 epochs using a MAP loss on the affine blocks, followed by 2 epochs using an unregularized ELBO loss on the affine blocks, followed by 50 epochs of reverse KL loss (see appendix E.4 for the training strategy, total 102 epochs).

For the MSHBM example in toy dimensions (see appendix E.3):

- ADAVI: NF with 1 Affine block with diagonal scale, followed by 1 MAF with $[32]$ units. HE with embedding size 32, 2 modules with 2 SABs (4 heads), 1 PMA (seed size 1), 1 SAB and 1 linear unit each. Minibatch size 32, 32 theta draws per X point (see appendix C.5), Adam ($10^{-3}$), 5 epochs using a MAP loss on the affine blocks, followed by 1 epoch using an unregularized ELBO loss on the affine blocks, followed by 5 epochs of reverse KL loss (see appendix E.4 for the training strategy, total 11 epochs).

For the MSHBM example in (see section 3.5):

- MF-VI: variational distribution is

$$
\begin{aligned}
q = {} & \mathcal{L} \circ \mathcal{N}([\mu_1^g, \ldots, \mu_L^g]; \text{mean=}V_{(L,D-1,)}, \text{std=} \mathrm{Exp}(V_{(L,)})) \\
& \times \mathrm{Exp} \circ \mathcal{N}([\epsilon_1, \ldots, \epsilon_L]; \text{mean=}V_{(L,)}, \text{std=} \mathrm{Exp}(V_{(L,)})) \\
& \times \mathcal{L} \circ \mathcal{N}([\mu_1^s, \ldots, \mu_L^s]; \text{mean=}V_{(S,L,D-1,)}, \text{std=} \mathrm{Exp}(V_{(L,)})) \\
& \times \mathrm{Exp} \circ \mathcal{N}([\sigma_1, \ldots, \sigma_L]; \text{mean=}V_{(L,)}, \text{std=} \mathrm{Exp}(V_{(L,)})) \\
& \times \mathcal{L} \circ \mathcal{N}([\mu_1^{st}, \ldots, \mu_L^{st}]; \text{mean=}V_{(S,T,L,D-1,)}, \text{std=} \mathrm{Exp}(V_{(L,)})) \\
& \times \mathrm{Exp} \circ \mathcal{N}(\kappa; \text{mean=}V_{(1,)}, \text{std=} \mathrm{Exp}(V_{(1,)})) \\
& \times \mathrm{Dirichlet}(\Pi; \text{concentration=}V_{(L,)})
\end{aligned}
$$

We used the Adam optimizer with a learning rate of $10^{-2}$. The optimization was ran for $10,000$ steps, with a sample size of $32$.

- ADAVI: NF with 1 Affine block with diagonal scale, followed by 1 MAF with $[1024]$ units. HE with embedding size 1024, 3 modules with 1 SABs (4 heads), 1 PMA (seed size 1), 1 SAB and 1 linear unit each. Minibatch size 1, 4 theta draws per X point (see appendix C.5), Adam $(10^{-3})$, 1000 epochs using a MAP loss on the affine blocks, followed by 1000 epochs using an unregularized ELBO loss on the affine blocks, followed by 1000 epochs of reverse KL loss (see appendix E.4 for the training strategy, total 3000 epochs).

