# OpenReview forum: "ADAVI: Automatic Dual Amortized Variational Inference Applied To Pyramidal Bayesian Models"
_ICLR.cc/2022/Conference — ICLR 2022 Poster_

### Official Review · Reviewer_KUup · 2021-10-28

**Correctness:** 4
**Technical Novelty And Significance:** 4
**Empirical Novelty And Significance:** Not applicable
**Recommendation:** 6
**Confidence:** 2

**Main Review:**

Overall, I am not an expert in HBMs and Neuroscience, but based on the introduction in this work, the proposed method is meaningful to the
field of Bayesian modelling for the neuroimaging, as it can estimate posterior distributions for a generative model of pyramidal structure, which is difficult to achieve. Authors provide sufficient experiments to support their claim in terms of both the inference quality and the speed. So I hold a positive attitude toward this paper. The paper is not very reader-friendly, so it would be better that authors can provide the introduction of some basic concepts and more related works in their final version to improve the readability of their paper.

**Summary Of The Paper:**

This paper proposes to derive an automatic methodology that takes as input a generative HBM and generates a dual variational family able to perform amortized parameter inference. The proposed method can be used to the context of  pyramidally structured data with good inference quality and a favorable training time.

**Summary Of The Review:**

Please refer to the Main Review section.

---

> ### Author Response · Authors · 2021-11-13
> **We answer to your query requiring additional background context**
>
> Dear reviewer,
>
> Thank you for the time invested in this review, and your kind view over our work. We apologize if you found our manuscript not to be reader-friendly. As you may come from a broader audience than our specific field, we value your point of view and would be glad to build upon it to improve our wide understandability.
>
> It is however unclear to us which precise points you found insufficiently contextualized. Which basic concepts do you feel our manuscript is lacking (approximate inference, Variational Inference, graphical models and plates, flow-based methods, amortization, etc...)?
>
> Subject to our space constraints, we would be glad to include more information in our introduction and/or methods sections.
>
> Thank you again for your help improving this manuscript,
> Best regards,
> The authors

---

### Official Review · Reviewer_bt4m · 2021-11-01

**Correctness:** 3
**Technical Novelty And Significance:** 3
**Empirical Novelty And Significance:** 2
**Recommendation:** 5
**Confidence:** 3

**Main Review:**

Strength:

- A good motivation: Extending the applications of simulation-based inference and structured variational inference to high-dimensional data settings (such as neuroimaging data) is interesting.

Weaknesses:

- The proposed ADAVI method only applies to a pyramidal class of Bayesian networks in which dependency structure follows a pyramidal graph. The authors need to motivate in the text that this class of problems covers most cases in the target applications in neuroimaging studies.
- The proposed method in its ultimate performance becomes similar to a simple mean-field approximation. The authors claim the proposed method instead provides more expressivity while failing to show this point in the experiments.
- The performance improvements (in terms of quality of inference and inference time) compared to other alternatives remain marginal (results in Table 1,2 and Figure 3).
- Despite the promising abstract and introduction setting a high expectation on the high-dimensionality of target problems (thousands of brain locations), the experiment on neuroimaging data is merely conducted on a relatively small dataset with 30 subjects and 1483 measures. This barely fits with the requirements in the field. These days we need methods that can deal with considerably larger datasets consisting of thousands of subjects with thousands of measures (for example UKBiobank). Furthermore, it would be nice to also compare quantitatively the time complexity and inference quality of ADAVI with other alternatives when applied to neuroimaging data.

Minor suggestions, comments, and questions:
- It may be nice to motivate the importance of HBR in the neuroimaging context. Why it is important to be able to handle the hierarchically organized data in the neuroscience context?
- Section 2.1: Direct Acyclic Graphs => Directed Acyclic Graphs
- Throughout the text: fig. => Fig.
- Section 2.1: A RV's hierarchy => An RV's hierarchy
- What do you mean by "symmetry" in "... exploiting the symmetry induced by the plates ..."?
- Section 3: "a a hierarchy" => "a hierarchy"
- Is the curves in Fig. 2a derived empirically? Then why the wall time for ADAVI remains fixed until $10^5$ examples? Please explain.
- While the main arguments are around the lower computational complexity of the proposed method, it is very difficult to judge the time improvements when diverse hardware are used in a heterogeneous cluster of computers (for example two types of CPU). Would be nice to fix the hardware setup to ensure fairness and reliability of comparisons.
- I suggest moving sections B.2 and B.3 to the main text (possibly to the final discussion) as they contain important information about the presented method.
- Would be great also to see some results in section E.5 in the main text (on how the individual network can deviate from the population network). That would be of high interest to the audience from the neuroscience community.


**Summary Of The Paper:**

This manuscript proposes an amortized variational inference to produce a dual variational family for hierarchical Bayesian models (HBM) that can be represented as pyramidal Bayesian models. The presented method exploits the exchangeability of parameters in HBM to reduce its parametrization for faster inference on high-dimensional data such as neuroimaging. The authors compare empirically the proposed method with several amortized and non-amortized alternatives and on several experimental data in terms of the size of parametrization, inference time, and quality of inferences.

**Summary Of The Review:**

Despite a nice motivation, the empirical results show a minor improvement over the already existing methods. The results are also weakly presented for the target problem in the neuroimaging context (small data, no quantitative comparison, lack of discussion). The method is not tested on very high dimensional settings as they are in the neuroimaging context.

---

> ### Author Response · Authors · 2021-11-13
> **We answer to your various comments and ask for precisions should you require additional quantitative experiments 1/4**
>
> Dear reviewer,
>
> Thank you for the time you invested in this review. We get the sense that you are particularly interested in our Neuroimaging application, and regret that you are not convinced of the interest of our method in this setup. We apologize if we created expectations for you that weren't met by our manuscript, but we argue that our contribution is of scientific value, and is a measurable step in the directions of the applications you have in mind.
>
> We'll try and answer to your concerns point by point, and stand at your disposal should you require experimental results that would convince you of the quality of our work:
> * **A - Motivating the class of Pyramidal HBMs:**
>     * TLDR *We acknowledge this class of models to be constraining, but argue that it makes our contribution much easier to understand*
>     * As indicated in section 2.1, pyramidal HBMs only prevent the presence of an observed RV at a hierarchy > 0, and of colliding plates:
>         * Regarding the first limitation, we argue that Pyramidal models cover the usual use-case -notably in population studies, and in Neuroscience as you point out- where observations are only made at the subject level, so at hierarchy 0. This confirms your opinion.
>         * We acknowledge the second limitation to be more constraining, notably preventing us from treating examples such as the Student-Course example (Koller 2009). But we believe that pyramidal model cover the classical setup of nested group structures (all other Student examples from Koller).
>     * Extending our method to a broader class of models is feasible and the subject of future work, but complicates much the explanation of our method. Indeed, this explanation is much facilitated by the notion of hierarchy. We therefore believe the current contribution to be a necessary intermediate step, and of scientific value.
>     * **Proposal plan of action:** better motivate pyramidal models in section 2.1
> * **B - Comparison to Mean Field variational inference**:
>     * TLDR *We differ significantly from MF-VI in at least 2 non-trivial aspects: amortization and expressivity in non-conjugate cases*
>     * You argue that our method asymptotically results in a classical Mean Field Variational Inference (MF-VI) scheme, and shares the same expressivity limitations. We believe that this statement can be nuanced on several aspects.
>     * First, our method is amortized, which makes it qualitatively different from a Mean-Field VI. Once trained, ADAVI can readily perform inference on any new data point, whereas MF-VI needs to be optimized again. Even matching MF-VI's performance but in the amortized setup would already be a non-trivial result.
>     * Second, the parametric form of MF-VI is fixed, and usually taken as the one of the prior. As our experiment in section 3.2 (NC) demonstrates, this limits the expressivity of MF-VI in non-conjugate cases. The fact that MF-VI stands as the best method in the GRE and GM cases in terms of posterior quality results from a careful construction of those examples on our side, and is not representative of the performance of MF-VI in the general case.
>     * As further explained in our answer to reviewer 4eiB (point C), our Mean Field approximation means that we don't model statistical dependencies between RV blocks over which we fit Normalizing Flows. The combination of several -even independent- NF blocks is non-trivial, and wasn't proposed before Cascading Flows (Ambrogioni et al. 2021), developed during the same period as our work.
>     * **Proposal plan of action:** our relationship to MF-VI should be stated more clearly, notably bringing section B2 to the main text.

---

> > ### Author Response · Authors · 2021-11-13
> > **2/4**
> >
> > * **C - Only marginal improvements compared to SOTA techniques in terms of inference time and posterior quality**:
> >     * TLDR *Our main claim is to match the performance level of SOTA techniques, keeping a more favorable parameterization adapted to larger data regimes, which we argue to demonstrate unequivocally*
> >     * Please also refer to the point F from our answer to reviewer bbMw
> >     * We argue that the necessary evidence to prove our claims is present in the current manuscript, but tat we failed to put it forward in a clear manner.
> >     * Our main claim is not to improve upon the posterior quality or training time of existing flow-based techniques, but to have a parsimonious parameterization that allows us to perform inference in regimes not reachable for those, such as for our Neuroimaging experiment.
> >     * In the GRE experiment, we argue that we match the performance level of SOTA techniques, but keeping a constant parameterization when the plate cardinalities grows exponentially, which is a non-trivial result.
> >     * In the GM experiment (Table 1), we also clearly stand as a superior amortized technique in the sense that we are on par with TLSF-A in terms of posterior quality, with a 2-fold lower training time. We also clearly improve upon the SOTA technique CF-A, showing that the adaptation of our variational family to the repeated structure of the generative HBM is also useful from a pure posterior quality point of view. This improvement can be visualized in the sup. mat. Figure D.3.
> >     * The improvement of the training time is a complex subject in itself, and is the subject of future work. We don't claim to drastically improve it.
> >     * **Proposal action plan:** we need to improve on the presentation of our experiments for our improvements to be more legible. We propose ton include a whole section in our experiments dedicated to this
> > * **D - Reaching larger parameter regimes such as the one from the UK Biobank:**
> >     * TLDR *We share your interest for those applications, and propose a method theoretically adapted to those, but limited as of today in its implementation*
> >     * We think to share your interest for extremely large regimes such as the one from the UK Biobank. We argue that we take a meaningful step in that direction.
> >     * Concretely, our Neuroimaging experiment features $L=2$ connectivity networks, represented using $D=1483$ connectivity vectors. We consider a cohort of $S=30$ subjects with $T=4$ measurement sessions per subject. This brings our total latent parameter size to (neglecting low-size latent parameters) $LD + LDS + LDST \simeq 0.4$ million parameters. Given the $N=314$ vertices in Broca's area, the observed data is of size $DSTN = 55$ million values. This data regime is unreachable for modern flow-based techniques, and is the one advertised in our abstract.
> >     * This data regime therefore calls for carefully-designed, problem-specific methods such as the original derivation from Kong et al. (2018). This analytical barrier to entry would make such HBMs impossible to use in practice for everyday Neuroimaging practitioners. We on the contrary deliver to those practitioners a technique automatic, efficient, and that provides full posterior distributions instead of point estimates.
> >     * Should a practitioner want to change a detail in the HBM, she wouldn't need to re-derive analytically a manual point inference methods: she could re-use our method seamlessly.
> >     * Our parameterization is furthermore independent from the plate cardinalities. This means that we could theoretically apply our method to any number of subjects S, sessions T, vertices N, with *the same* number of model weights. For instance to perform full-cortex inference on the entirety of the HCP dataset. In practice however, training in those regimes creates too big activation tensors, and blows up computer memory. This is an engineering, and not a theoretical limitation. We take this limit seriously and are investing efforts in developing *sharding* and training method that could help us tackle those regimes.
> >     * In conclusion, we provide the Neuroimaging community with an automatic method that is theoretically suited for very large dataset inference. We deem this to be a scientific contribution of interest, that with the proper engineering effort could be converted into a tool useful for practitioners
> >     * **Proposal plan of action:** we propose to better motivate our method in section 3.5, while presenting our Neuroimaging results.

---

> > > ### Author Response · Authors · 2021-11-13
> > > **3/4**
> > >
> > > * **E - Comparing to other methods in the Neuroimaging context:**
> > >     * TLDR *This experiment was not meant to be a benchmark, but we can attempt to fit a MF-VI to this example and compare both method's ELBOs*
> > >     * This is a valid experimental concern, but difficult to put in place in practice. As you point out, our intent was to prove the utility of our method on this example, and not to use it as a benchmark, preferring to do so on more classical and better-understood examples like GRE and GM.
> > >     * First, Flow-based methods cannot be applied in this regime due to the parameter sizes presented above. This point is related to our main claim.
> > >     * With a careful design, MF-VI could probably be applied in practice. Following our discussion with reviewer 4eiB (point D), we could in this context then compare the method's respective ELBOs. However in this context there is most certainly an approximation gap for both methods, so we wouldn't be able to diagnose the actual KL divergence to the true posterior.
> > >     * **Proposal plan of action:** subject to our time constraints, we can design a MF-VI method on the Neuromimaging example and add the comparison results to our supplemental material
> > > * **F - Moving the supplemental discussion to the main text:**
> > >     * TLDR *There seems to be consensus that B.2 should be included in the main text, but we argue that B.3 is at the right position in our supplemental material*
> > >     * The supplemental sections B.2 and B.3 respectively discuss the Mean Field approximation and the extension of our method to a broader class of simulators.
> > >     * You share your point of view over section B.2 with reviewers 4eiB and bbMw, and we thus plan to move it to the main text
> > >     * However, we believe that section B.3 shouldn't be moved to our main text:
> > >         * First, because it pushes us further from the VI framework (based on the availability of a generative HBM density $p$). We believe that the inclusion of B.3 in our main text could confuse readers
> > >         * Second, though we argue that our methodology could be extended to likelihood-free setups, we did not at this stage experiment on this possibility. We furthermore do not plan to do so in future work. One reason for this is that the API suited for generic simulators would probably be very complex, and user-friendliness is a key concern for us. Another reason is that our experiments show forward-KL based training -necessary for this extension- to be less adapted to our data regimes.
> > >     * Therefore, we believe that the presence of B.3 in our supplemental material is useful to open up research possibilities, but that the space cost to include it in our main text is not worth the benefits.
> > >     * **Proposal plan of action:** we propose to move B.2 to our main text, but not B.3
> > > * **G - Weak presentation of the Neuroimaging example, inclusion of subject-level results:**
> > >     * TLDR *Those results are interesting, but not for a sufficiently large portion of the ICLR readers to be included in the main text*
> > >     * We are glad to share your interest in our Neuromimaging results. We regret not to be able to put more results related to Neuromimaging in our main text.
> > >     * But we believe in this context ICLR to be a rather methodological conference, and that there wouldn't be a broad interest in the ICLR community to see more Neuroimaging results.
> > >     * Furthermore, the main text's space is limited, and the inclusion of more Neuroimaging results would necessary be made at the expense of other results validating our methodology
> > >     * Lastly, we believe that the main text is clear in pointing out interested readers to our supplemental material for additional results.
> > >     * We actually believe that a proper publication in a Neuroimaging journal would be a better stage for more complete results.
> > >     * **Proposal plan of action:** we propose to keep the main text as is
> > > * **H - Motivating HBMs in Neuroimaging:**
> > >     * See point B in our answer to reviewer 4eiB
> > > * **I - "...exploiting the symmetry introduced by the plates..."**
> > >     * We apologize for the slight abuse of language that makes our writing confusing
> > >     * In this context, "symmetry" refers to the permutation invariance for the ground RVs across a plate, or equivalently the Bayesian exchangeability in the HBM. "Symmetry" appealed to us as a generic term to designate a repeated structure, or pattern, in the HBM
> > >     * **Proposal plan of action:** we propose to rewrite the sentence, replacing "symmetry" with "exchangeability"

---

> > > > ### Author Response · Authors · 2021-11-13
> > > > **4/4**
> > > >
> > > > * **J - Explain Figure 2:**
> > > >     * TLDR *Our figure is confusing, and we propose to update the profile for ADAVI to an affine one*
> > > >     * Figure 2 is related to the total wall time to perform inference over a number of different data points. The result may appear as trivial for experienced readers, but we believe amortization to be an important concept throughout our experiments. Therefore, we deemed useful to represent this notion graphically.
> > > >     * Our curves are indeed obtained empirically.
> > > >     * We propose to clarify how we obtained this figure:
> > > >         * MF-VI is a non-amortized technique, ran over a fixed number of optimization steps for any new data point presented. As such, there is no upfront training time for the method, and the inference time is fixed for any data point. Therefore, the cumulative time to infer for several examples is linear with respect to the number of examples
> > > >         * ADAVI is an amortized technique. As such, we first train over the generative HBM, which represents an upfront training time. But once the model is trained, inference over a new data point requires no further optimization, and is quasi-instantaneous. We believe our mistake was to further infer over batches of examples, as permitted by our architecture. As a result, the inference time is constant until we fill the GPU memory, after which it becomes linear. This explain the 2-stage profile of our inference time
> > > >     * We believe that our presentation mixes theoretical points and implementation details, and thus becomes confusing. We propose to remove the batched evaluation for our method ADAVI, and to thus to compare a linear profile (MF-VI) with an affine one (ADAVI)
> > > >     * **Proposal plan of action:** we propose to change the figure as described
> > > > * **K - Hardware discrepancy:**
> > > >     * We did not keep any traceability on precisely which experiments were performed on which hardware. Due to our cluster scheduling and priorities however, we can assume that most of our experiments were performed on AMD EPYC 7742 64-Core Processor (512Mb RAM) CPUs and NVIDIA Quadro RTX 6000 (22Gb) GPUs
> > > >
> > > > Thank you again for your help improving this manuscript,
> > > > Best regards,
> > > > The authors

---

### Official Review · Reviewer_4eiB · 2021-11-02

**Correctness:** 4
**Technical Novelty And Significance:** 2
**Empirical Novelty And Significance:** Not applicable
**Recommendation:** 6
**Confidence:** 5

**Main Review:**

Strengths:
- The method deviates from black-box simulators and instead exploits the hierarchical nature of many forward processes to scale to high-dimensional parameter spaces.
- Experiments are detailed, and results are discussed carefully. ADAVI shows good performance, in particular against (S)NPE, but non-amortized methods are shown to achieve higher ELBO.
- The supplementary materials are thorough and include helpful experimental details as well as further discussions.

Weaknesses:
- Population studies are taken as an example of large HBMs with millions of parameters. I believe it would be fair to mention that oftentimes only a few of those many parameters are actual of interest for scientific inquiry. In this case, most of the parameters are latent variables for which no explicit estimation is necessary. Would you argue that ADAVI is still relevant in this case? [I would say it may still be relevant, but I would be curious in having your opinion.] Could you also better motivate in which scientific cases inference over millions of parameters is strictly necessary?
- In Section 2.3, the variational distribution is defined as a mean-field approximation. Can you comment on the constraints that result from this assumption?
- Results are discussed in terms of the ELBO only. The quality of the approximate posteriors is never diagnosed explicitly (at least as far as I can see).
- I would appreciate some comments on the impact of the size of the encoding space produced by the SetTransformers. How shall one set this size? Should it be the same across all levels?

Small remarks:
- Avoid footnotes if you can.
- Replace "fig. X" with "Fig. X".

**Summary Of The Paper:**

This work introduces ADAVI, an approximate inference algorithm for hierarchical Bayesian models (HBMs). The approach is similar to NPE from simulation-based inference but exploits the hierarchical structure of the forward model to generate an efficient variational family automatically. Experiments demonstrate the applicability of the method on HBMs of increasing complexity, including a challenging neuroimaging model. Results indicate good performance against other amortized methods.

**Summary Of The Review:**

This paper addresses a common problem of simulation-based inference and proposes a sound and efficient solution to enable inference in high-dimensional parameter spaces. Experiments show convincing results.

My main issue is the lack of quality checks of the approximate posteriors produced by ADAVI. This should matter the most, in my opinion, well above efficiency, wall-clock times, and the number of parameters. If the approximate posteriors are wrong, none of those matter. For now, I do not recommend this paper for acceptance, but I will be pleased to change and increase my evaluation if the authors can present diagnostics of the resulting approximate posteriors.

---

> ### Author Response · Authors · 2021-11-13
> **We answer to your various comments and ask for precisions should you require additional quantitative experiments 1/3**
>
> Dear reviewer,
>
> Thank you for the time you invested in reviewing this paper, and for the interest you have in our method.
>
> For now, we attempt to provide you with answers to the many interesting points you raised. If necessary, our objective is to provide you later on with a more quantitative answer, but we would appreciate any help you could provide us to refine the *posterior diagnosis* you want to find in our manuscript.
>
> Regarding your various remarks:
> * **A - Relevance of ADAVI in cases where some latent parameters are not of interest**. To our knowledge:
>     * TLDR *Ignoring some latent parameters would be possible switching to a forward KL loss, at the cost of a less effective training.*
>     * *Reverse KL case:* This is a good experimental concern: sometimes one could only be interested in the posterior mean of a Gaussian and not so much in the posterior variance. Given a fixed generative HBM with latent parameters $\{\theta_1, \Theta\}$ we cannot however "ignore" $\theta_1$. Let's imagine we'd only provide with a variational distribution $q(\Theta)$ instead of $q(\theta_1, \Theta)$. To keep the ELBO normalized, in its computation one would needs to compute $p(X, \Theta)$ instead of the known $p(X, \theta_1, \Theta)$. So one would need to marginalize $\theta_1$ in $p$, which isn't easy a priori, even with a Monte Carlo scheme. As far as we know, in that case it would probably be better to modify the generative model and to remove $\theta_1$ altogether from the problem.
>     * *Forward KL case:* in the forward KL case, the evaluation of $p$ does not appear. So one could actually "ignore" $\theta_1$ -that is to say not provide any estimator for the posterior distribution of $\theta_1$. Effectively, the effect of $\theta_1$ would be marginalized. However, as our experiments in the GRE and GM cases underline, forward KL training seems to fair worse than reverse KL training in high plate cardinalities or for a complex problem. So our ability to ignore certain latent parameters would come to a cost.
>     * **Proposal plan of action:** we propose to include a more detailed discussion of this point in our supplemental material, section B.
> * **B - Cases in which inference over a million hyperparameters is strictly necessary**:
>     * TLDR *It is our understanding that those many parameters emerge from the combination of a hierarchical structure with high-dimensionality data points.*
>     * We think our Neuroimaging example (Kong et al. 2018) to be a good illustration:
>         * *Population parameters:* in Neuroscience measurements are costly, there is a weak signal-to-noise ratio, and the data is very high-dimensional. This calls for methods with the ability to combine data from multiple sources to gain statistical power. In the example from Kong et al. (2018), this comes with a cohort of multiple subjects, and multiple measurement sessions per subject. Those large plates combined with large data point dimensionality quickly build up to millions of parameters.
>         * *Subject and session-level parameters:* It is furthermore of interest in Neuroscience to compare for instance the deviation of an individual's brain connectivity from the population. Instead of the typical local vs global parameters paradigm (HVI, Ranganath et al. 2016), this calls for a nested hierarchical structure, which naturally multiplies the number of latent parameters;
>     * Other examples are the author-topic models (SVI, Hoffman et al. 2013), or experiments one could try and perform on the UK Biobank as reviewer bt4m points out.
>     * **Proposal plan of action:** we propose to better motivate our use-case at the beginning of section 2.1

---

> > ### Author Response · Authors · 2021-11-13
> > **2/3**
> >
> > * **C - Commenting on the Mean Field (MF) approximation**:
> >     * TLDR *The Mean Field approximation streamlines our contribution but limits our expressivity, and is to be modified in future work.*
> >     * As pointed out in section B.2, this is indeed a main limitation of our method. But this is a readily modifiable implementation choice, made to streamline our contribution (SSVI, HVI and CF stand as possible designs to correct this).
> >     * We argue that we actually provide an intermediate between a vanilla Mean Field (every latent parameter is independent) and a full statistical dependency modeling (such as for TLSF). Precisely, our constraint is that we assume the statistical independence of the latent parameter blocks over which we fit Normalizing Flows. Inside a given block however we model all the statistical dependencies.
> >     * Going further, one could concatenate the parameter spaces of template RVs sharing the same hierarchy to model the statistical dependencies between those RVs. With the current design, remain 2 types of statistical dependencies that cannot be modeled: the dependencies between RVs of different hierarchy, and the dependencies between ground RVs *across* a given plate.
> >     * The Mean Field approximation limits our expressivity, but we think it makes our contribution easier to delineate and to understand for the reader. We will modify this approximation in further work.
> >     * **Proposal plan of action**: we propose to include a more detailed discussion of this point in section B2. B.2 is also meant to be brought to the main text.
> > * **D - Explicitely diagnosing the posterior quality:**
> >     * TLDR *We built our GRE and GM experiments so as to consider the proximity to the MF-VI ELBO as a good diagnosis for posterior quality.*
> >     * We understand this is your main concern, and we are willing to produce any results that would convince you on that matter. However we believe this diagnosis to be a complicated point to address, and would appreciate if you could provide us with more information, or even a precise metric you have in mind.
> >     * The choice of a valid quality metric is an open subject, as the review from Lueckmann et al. (2021) points out. As far as we know, the absence of a ground truth in inference means there is no perfect metric, especially in the GM example featuring the label switching issue. This choice of metric complicated the most the redaction of this paper, all the more since we aim at comparing methods from different literature, coming with different historical metrics (VI, SBI, ...)
> >     * We propose to detail our reasoning. To our understanding, a valid metric would be the KL divergence between the posterior $p(\theta | X)$ and the variational distribution $q(\theta)$. We cannot however compute this metric due to the fact that $p(\theta | X)$ is unknown, yet:
> >         * The ELBO relates to this divergence through $\text{KL}(q(\theta) || p(\theta | X)) = \log p(X) - \text{ELBO}(q)$
> >         * We specifically built the GRE and GM examples so that MF-VI has no approximation gap due to conjugacy. So the ELBO values of MF-VI are close to the highest ones that can be achieved for those examples. Assuming the KL divergence in the case of MF-VI is close to 0, the variations of the ELBO are then only due to the evidence of the data $p(X)$, that is constant for all inference methods. A method with the same ELBO as MF-VI would therefore have a similarly very low KL divergence.
> >         * As a consequence of the two previous points, closeness to the MF-VI ELBO seems to be a valid proxy to a good posterior quality
> >         * As such, MF-VI has a particular status in our benchmark, that should be underlined graphically in our tables and figures
> >     * If in your opinion another metric would be more suited, we would be ready and willing to compute it. However as the sup. mat. Table 2 underlines, computation over the GRE example takes several weeks, and we could only provide you with preliminary results (low plate cardinality) that would be complemented provided our draft passes this rebuttal stage.
> >     * **Proposal plan of action:** we argue that we do diagnose the quality of our posterior. We propose to update our figures to better show the role of MF-VI as a proxy to the ground truth posterior. Albeit our time constraints, we are willing to produce any alternative experiments that would otherwise convince you, but would appreciate your guidance on that matter.

---

> > > ### Author Response · Authors · 2021-11-13
> > > **3/3**
> > >
> > > * **E - Encoding size for Set Transformers**:
> > >     * TLDR *This is an important hyperparameter that we kept constant across hierarchies, and roughly increasing with the complexity of the inference problem*
> > >     * Hierarchical Encoder embedding size is an important hyper-parameter for our method. For a single network, its importance is already studied in the original paper from Lee et al. (2019). As a consequence, we did not study experimentally its effect, and fell back onto a conservative approach.
> > >     * Set Transformers collect summary statistics at a given hierarchy. The summary statistics at a high hierarchy $h_+$ are computed using lower-hierarchy $h_-$ summary statistics. Intuitively, it seems important not to bottleneck the information flow at the $h_-$ level with a too low embedding size: at this level, information useful for the inference of both the RVs at hierarchy $h_-$ but also at level $h_+$ needs to be gathered.
> > >     * Intuitively, the size of an encoding vector, shared by all template RVs of a given hierarchy, should be increased with the number of RVs depending on it, the complexity of the inference task at hand, and the dimensionality of the latent RVs.
> > >     * In our experiments, we only studied three different sizes on synthetic examples: 8 (NC, GRE), 16 (GM) and 32 (MSHBM), kept constant at all hierarchies. Those numbers were roughly dependent on the supposed complexity of the inference problem at hand. For the Neuroimaging experiment, the embedding size was 1024.
> > >     * It would indeed be interesting in further works to develop schemes to vary the embedding size across hierarchies, and to have principled ways to set its value.
> > >     * **Proposal plan of action:** we propose to integrate this discussion in our supplemental material B.
> > >
> > > Thank you again for your help improving this manuscript,
> > > Best regards,
> > > The authors

---

> > > > ### Author Response · Authors · 2021-11-13
> > > > **Removal of non-amortized techniques from the main text**
> > > >
> > > > * Following the various helpful remarks from this review process, we believe that the presence of amortized methods in our benchmarks complicates the understanding of our manuscript while not being of essential use
> > > > * We radically propose to move:
> > > >     * section 3.1
> > > >     * the non-amortized results from methods CF-NA, TLSF-NA and SNPE-C in sections 3.3 and 3.4
> > > > * ... to our supplemental material, therefore gaining some needed space in the manuscript, and making our experiments more understandable
> > > > * We would keep only MF-VI in our benchmark, clearly delineated as a proxy for the ground truth posterior
> > > > * **We would appreciate the advice of the reviewing committee on that matter**

---

> > > > ### Comment · Reviewer_4eiB · 2021-11-19
> > > > **Thanks**
> > > >
> > > > Dear authors,
> > > >
> > > > Thanks a lot for your detailed answers and for your concrete proposals to further improve the manuscript. I am happy with all the changes that you suggest. I believe they will greatly improve the quality of your work.
> > > >
> > > > Regarding D, I appreciate your point that the MF-VI ELBO can be used as a good proxy of the posterior quality.
> > > >
> > > > I have increased my evaluation from 5 to 6.

---

### Official Review · Reviewer_bbMw · 2021-11-02

**Correctness:** 3
**Technical Novelty And Significance:** 3
**Empirical Novelty And Significance:** 2
**Recommendation:** 5
**Confidence:** 3

**Main Review:**

# Strengths
The motivation for the work is clear and it's likely to be of broad interest. Factorizing the approximation network to match the structure of the model is interesting and novel to my knowledge, and the experiments build up from synthetic toy examples to a real-world problem.

# Opportunities for improvement

## Overall framing and relation to broader approximate inference
The overall framing of the paper starts from the perspective of improving the tractability of normalizing flows for posterior approximation rather than improving approximate inference, and takes its aim primarily at flow-based methods (notably, cascading flows) rather than approximate inference more broadly. I think broader engagement with the VI literature may improve the paper, including particularly some mention of the approximation vs amortization gap. This then situates the MF VI baseline as not just a "common practice" method but (as far as I can tell) the upper bound on performance of the remaining methods, which share its approximation gap but add an amortization performance gap. I'm not sure I understand the non-conjugate results in this context, though -- is it because the link function is applied to the flow approximation but not in the MF context? Relatedly, I think the discussion from supplement B2 is worth including in the main paper -- the HVM and SSVI approaches among others seem like a natural complement to the present work. Making these connections seems more important than engaging with SBI and likelihood-free inference -- I don't think likelihood-free approaches are ever competitive with likelihood-based approaches if likelihoods are available (nor is this their intent), so I'm not sure why they are an important baseline or particularly relevant to the current work. Finally, as a minor point, I wonder if the more typical reader here would come from the VI literature, for whom leading with the ELBO (rather than Reverse KL) framing would be more intuitive.

## Pushing the synthetic examples to regimes where ADAVI wins
None of the examples actually show that ADAVI is a superior approach: in the NC case it is roughly equivalent in performance to CF. In GM ADAVI may achieve the best ELBO out of the amortized methods, but takes an order of magnitude more time than just running MF-VI, so the benefit of amortization is not shown. In the gaussian random effects example (fig3) it's clear that ADAVI has favorable scaling, but even in the largest example MF-VI has fewer parameters and achieves higher accuracy in less time. Furthermore, I'm not sure if the selected metrics truly show the benefit of amortization: rather than showing optimization time for all the models, shouldn't the paper show the predictive inference time (and performance) for held out data for all the models? Then the amortized models should be dramatically faster by not requiring any optimization. Finally, I'm not sure why timings are not done on consistent hardware -- it seems like everything in the paper should run on both CPU and GPU, so picking one (or showing timing for both) would be better than trying to compare timings between two different sets of hardware.

## Including quantitative evaluation on the real problem
While the real-world results match qualitative patterns expected from prior work, the paper needs to do more to quantitatively show how ADAVI is indeed superior for these kinds of models. The best way to do this would be reporting cross-validated predictive loss, e.g. by holding out subjects, times, or connectivity measures, and comparing both the loss and runtimes against other models.

## Minor comments and typos

Section 2.2. could be clarified further. For example:
- Is the set transformer really elementwise? Or is it "elementwise" w.r.t. the leaf random variables in the graph?
- If the "contraction" operator isn't to tensor contraction, could a different word be used, considering that tensor notation is already used and "contraction" has a common meaning in that setting?
- $\mathcal{B}_h$ is defined twice above and below expr. 1.
- Both upper and lowercase $x^{i,j}$ is used -- is it the same or different?
- The meaning of the superscripts on $x$ is not explained (I think one is the plate index and the other is the datapoint index, but not sure).

In addition:
- The paper should clarify the full algorithm: is everything trained end-to-end? Or are the STs trained first, and then the model parameters?
- Section 3.2 could potentially pick a more practically-relevant non-conjugate example from the literature (there are plenty of examples in neuroscience, though I'm not sure if any have non-discrete latents).
- Fig2 and 3 are missing error bars/ribbons.
- $x \sim \mathrm{Mix}(\ldots)$ is not defined and not standard to my knowledge -- I'm assuming this is multinomial over mixture components given $\pi$ and each component is Gaussian?
- Expressions 9a-9b are not arranged in an intuitive way (definitions and priors are intermixed, everything is in one giant block). An annotated figure (even a plate diagram) would be better, and the specifics can be left to the supplement, especially since the paper doesn't give enough info to understand the model beyond the Kong et al. citation.



**Summary Of The Paper:**

The paper tackles approximate inference for hierarchical Bayesian models with fully nested structure. The specific approach taken is variational inference with a q-distribution that iteratively applies conditional normalizing flows to derive a hierarchical representation, and factorizes in a manner parallel to the generative model by reusing flow parameters, thus having a number of parameters that does not grow with cardinality of each plate. The benefits of the model are illustrated in a few synthetic experiments as well as in application to a human neuroimaging dataset.

**Summary Of The Review:**

Hierarchical graphical models of the sorts considered here are in wide usage by scientific practitioners, and automatic approximate inference methods that are efficient and accurate are of longstanding interest -- thus the motivation of the work is clear. The nature of the contribution is likewise clear. My primary concern is that the evaluation doesn't fully demonstrate the benefits of the approach (e.g. using predictive out-of-sample log-likelihoods rather than training ELBOs, including a quantitative evaluation of the real-world problem, providing consistent timings, pushing the scale until ADAVI dominates). Secondarily, I think the overall framing, clarity, and writing could be improved. I think the work is below the bar now and am rating accordingly, but I think my concerns should be sufficiently addressable in rebuttal for the work to be appropriate for the conference.

---

> ### Author Response · Authors · 2021-11-13
> **We answer to your various comments and ask for precisions should you require additional quantitative experiments 1/3**
>
> Dear reviewer,
>
> Thank you for the time invested in this review. We are glad to hear that you would consider our work appropriate for the conference, should we provide the work in this rebuttal necessary to prove the benefits of our approach.
>
> To do so, we would however respectfully ask for more guidance to define precisely which metrics and experiments would convince you. We wish to provide you with an appropriate rebuttal revision at a later stage, and would require some precision regarding various points:
> * **A - More general engagement with the approximate inference literature:**
>     * TLDR *We identify Monte Carlo methods and VAEs as lacking if we were to make a broader approximate inference contextualization*
>     * You are right to point out that our main point of comparison in this work is flow-based methods. Normalizing Flows are a relatively new topic, and we believe there is an interest in hierarchically combining ensembles of those, as our method or Cascading Flows (Ambrogioni et al. 2021) pertains.
>     * As for the more general engagement to the VI literature, what branches do you have in mind that should be mentioned? We acknowledge 2 such branches that we don't much refer to: Monte-Carlo methods (MCMC, SMC) and Variational Auto Encoders.
>     * **Proposal plan of action:** we propose to mention MC methods and VAEs in our introduction, and any other methods you would deem necessary.
> * **B - Approximation vs amortization gap:**
>     * To our knowledge, the approximation vs amortization gap concept has been mainly developed as part of the VAE literature (Cremer et al. 2018).
>     * Since the notion of amortization is central to our manuscript, we agree that analyzing our results using that framework is indeed very pertinent.
>     * **Proposal plan of action:** we propose to mention the approximation vs amortization gap both in our introduction and as part of our experimental results analysis
> * **C - Role of MF-VI in our manuscript:**
>     * TLDR *MF-VI does play the role of an upper bound for performance due to the careful design of our GRE and GM experiments. The NC case is a reminder that MF-VI however features an approximation gap in the general case*
>     * Please refer to the point D from our answer to reviewer 4eiB
>     * In order to get a good proxy to the KL divergence between our variational distribution and the true posterior, we carefully constructed our GRE and GM examples so that MF-VI wouldn't have any approximation gap. MF-VI thus has a particular status in our GRE & GM benchmarks, and is to play the role of the "best achievable ELBO"
>     * In the general case however, especially in non-conjugate cases, MF-VI would feature such an approximation gap. This in turn creates an interest for "universal" density estimators such as Normalizing Flows, that virtually wouldn't feature such a gap.
>     * The experiment NC (section 3.2) aims at illustrating that notion: in a non-conjugate case flow-based methods have a more favorable ELBO that a distribution with a fixed parametric form.
>     * As a consequence, methods such as NPE-C, TLSF, CF or ADAVI do *not* share the same approximation gap as MF-VI. Their amortized versions however do feature an amortization gap.
>     * **Proposal plan of action:** in accordance to our answer to reviewer 4eiB, we propose to visually underline the particular role that MF-VI plays in our experiments, and to reformulate section 3.2 so that to clarify that role
> * **D - Mean Field approximation:**
>     * Please refer to the point D from our answer to reviewer 4eiB
>     * You share your opinion about including section B.2 in our main text, which we plan to do
>     * Regarding the modeling of statistical dependencies in our variational family, you are right to point out designs such as HVM ans SSVI as helpful on that matter. We would also consider Cascading Flows or structured Normalizing Flows (Weilbach et al. 2020) as interesting designs
>     * This extension is subject of future work

---

> > ### Author Response · Authors · 2021-11-13
> > **2/3**
> >
> > * **E - Relevance of the SBI and likelihood-free setups:**
> >     * TLDR *There is a belief that likelihood-free methods are not meant to be competitive in the presence of a likelihood. We think that we have novelty in illustrating that belief quantitatively, and actually support a more nuanced viewpoint*
> >     * You underline the somewhat unfair comparison with likelihood-free methods in the context of an available likelihood. We share with you the belief that in the presence of a likelihood, likelihood-free methods should be less competitive.
> >     * However, to our knowledge, this lack of competitiveness has seldom been quantified. SBI methods have received much attention in the recent years, and we believe it is a fair question to ask what does a practitioner "loose" by resorting to those methods even in the presence of a likelihood?
> >     * Our GRE experiments are interesting in that matter: for $G=30$ NPE-C rivals TLSF-A in terms of posterior quality, but with a 10-fold lower training time. However NPE-C's performance degrades for $G=300$, hinting towards a worse scaling to high dimensionality. In the GM case, the performance of likelihood-free methods is however absolutely not competitive (probably due to the label switching issue).
> >     * As a conclusion, we believe likelihood-free methods have their place in our benchmark. As you point out, probably less so in our discussion.
> >     * **Proposal plan of action:** we propose to leave our experiments as is, but to reduce our focus on likelihood-free methods in our discussion
> > * **F - Interest of our method not being underlined experimentally**
> >     * TLDR *We argue that the necessary evidence is present in our manuscript, but should be explicitly compiled and presented to the reader*
> >     * You argue that no single experiment underlines the interest of our method. We apologize for the inconvenience, and argue that the interest of ADAVI is actually to be seen via a combination of all our experiments:
> >         * 3.1 underlines the qualitative difference between amortized and non-amortized techniques, and is meant to underline that the ELBO comparison between those 2 paradigms over individual examples is somewhat unfair (due to the amortization gap and the differences in inference time).
> >         * 3.2 underlines that MF-VI can lack expressivity in non-conjugate cases, and that its high performance in the GRE and GM examples is not representative of the general case.
> >         * The combination of 3.1 and 3.2 means that the relevant point of comparison for ADAVI are NPE-C, TLSF-A and CF-A.
> >         * 3.3 allows us to compare those different methods, with MF-VI playing the role of the upper bound for the ELBO. In this experiment, as per our main claim, we match the performance levels of other methods, but keep a constant parameterization with respect to plate cardinality
> >         * 3.4 then introduces a data regime over which no other flow-based -and therefore expressive in the generic case, contrary to MF-VI- can be applied
> >     * We argue that this combination unequivocally demonstrates the interest for our method. We however failed to make this point in our manuscript.
> >     * **Proposal plan of action:** we argue there is enough evidence in the present manuscript to prove our claims. We however acknowledge that our reasoning has to be made more explicit and propose to include a section in our experiments dedicated to linking all of our results. We stand at your disposal should our reasoning fail to convince you, and should you therefore require additional experimental results.
> > * **G - Showing inference times instead of optimization times:**
> >     * TLDR *Showing the inference time would shed a more positive light on amortized methods, but would prevent any meaningful quantitative comparison in-between those*
> >     * We believe that the comparison between amortized and non-amortized techniques is a complicated matter. Our attempt was to contribute quantitatively to this topic, but we argue that the difference between those methods is rather qualitative, and that the choice over one or another is dependent on the application at hand.
> >     * Our current choice to display the optimization time sheds a positive light on non-amortized techniques. Choosing to display instead as you propose the inference time would conversely shed a positive light on the amortized techniques.
> >     * We furthermore expect amortized methods to be drastically faster than non-amortized ones with respect to the inference time, but not to display any significant differences between each other. Yet amortized methods are the main focus of this manuscript, and it matters to us to compare amortized alternatives quantitatively.
> >     * As a consequence, we do not think that replacing the optimization time with the inference time would improve our manuscript.
> >     * **Proposal plan of action:** we propose not to modify our manuscript on that matter

---

> > > ### Author Response · Authors · 2021-11-13
> > > **3/3**
> > >
> > > * **H - Precision on the nature of the considered ELBO:**
> > >     * As part of this rebuttal, we realize that we lacked precision regarding our evaluation metric. In Table 1, Figure 3 and sup. mat. Table 2, ELBO refers:
> > >         * to the training ELBO for non-amortized methods, averaged over 20 examples
> > >         * to the validation ELBO over held-out data for amortized techniques, amortized over the same set of 20 points
> > > * **I - CPU vs GPU hardware inconsistency:**
> > >     * We agree that this point makes our experiments confusing.
> > >     * Our idea was to separate qualitatively the optimization time for amortized and non-amortized techniques.
> > >     * **Proposal plan of action:** we propose to remove GPU times altogether
> > > * **J - Quantitative assessment in the Neuroimaging experiment:**
> > >     * Please also refer to the point E of our answer to reviewer bt4m
> > >     * We precise that our method is never presented with any Neuroimaging data at training time, but only with synthetic points coming from the generative HBM. What's more, we only possess one such Neuroimaging point. As a consequence, we argue that the notion of held-out data is difficult to put in place in this context.
> > > * **Minor points:**
> > >     * **Leading with the ELBO and not the KL divergence:** we are satisfied with the current framing as we believe it to be easier to grasp for the SBI community, and useful for our section C.4
> > >     * **"Element-wise" Set Transformer**: as you point out, this is an abuse in notation and refers to the exchangeability of leaf nodes in the HBM
> > >     * **"Contraction" of tensors:** thank you for pointing out this abuse in notation referring to the summary of i.i.d data into a single encoding
> > >     * **$x^{i,j}$ notation:** $\mathbf{X} = [[x^{1,1}, x^{1,2}], [x^{2,1},x^{2,2}]]$ refers to the grounding of the RV template $\mathbf{X}$, whereas $X=[X^{1,1}, X^{1,2}], [X^{2,1},X^{2,2}]]$ refers to the slicing of the observed data point $X$. In both cases, index relate to successive plate indexing, in this case $\mathcal{P}_1$, $\mathcal{P}_2$ in that order. We'll try and come up with a clearer notation
> > >     * **ADAVI training:** the training is indeed performed end-to-end (both HE and the density estimators), we will precise it in section 2.3. On that note, transfer learning is also possible, as presented in section E.4
> > >     * **More relevant non-conjugate example:** in the continuous domain, we don't know of any famous example, and would appreciate your guidance on that matter
> > >     * **Error bars to Figure 2:** in our implementation optimization and inference times are constant, and there is no variance for that matter
> > >     * **Error bars for Figure 3:** we tried to add ribbons but those made the figure unreadable. We will try to experiment on that Figure's presentation
> > >     * **Mixture notation:** you are correct on your interpretation. We will properly introduce the notation
> > >     * **Equation 9 unreadable:** we will remove it altogether and move it to our supplemental material
> > >
> > > Thank you again for your help improving this manuscript,
> > >
> > > Best regards,
> > >
> > > The authors

---

### Author Response · Authors · 2021-11-20
**Rebuttal version uploaded 1/2**

Dear reviewers,

We thank you for the quality of your reviews, and the valuable time you have invested in writing them. We just uploaded a **new rebuttal version for our manuscript**, that we believe to address your comments according to our previous detailed answers. Thanks to your guidance, we believe this new draft to be of greater quality, and to put forward more clearly our main claims. To help you in your review, **the main changes in our draft have been colored in blue**.

## General comment

Regarding your reviews, we believe the committee to have a **positive view on our motivation and our method**: we want to bring the expressivity of flow-based methods to population studies featuring a large number of latent parameters. As such, **we don't claim to improve on the posterior quality nor on the amortization time compared to existing methods, but to have a parameterization invariant to plate cardinalities**. This parsimonious parameterization in turn allows us to tackle population studies effectively.

We believe this main claim to be original, in the sense that most manuscripts in approximate inference focus on improving the quality of the variational posterior. As such, **the evidence to prove our claims is to be seen in a combination of all our experiments**:
* sections 3.1 and 3.2 underline desirable properties that the common practice method MF-VI lacks: **amortization** and **expressivity in non-conjugate cases**. The relevant benchmark for our method are flow-based methods, and we compare ourselves to those methods in experiments where we can use MF-VI as a proxy to the ground truth posterior;
* in section 3.3, **we match the performance of benchmarked flow-based methods** in terms of inference quality and amortization time. But we do so with **orders of magnitude less weights**, as explained theoretically in section 2.2;
* as such, **our method can be applied to parameter regimes unreachable for existing flow-based methods**, that incur a computer memory blow-up: a neuroimaging experiment featuring half a million latent parameters in section 3.5;
* even if this isn't our claim, **the inference quality of our amortized method is actually competitive**, and is for instance superior to the one of a SOTA method (Cascading Flows, Ambrogioni et al. 2021) on a classical yet complex example: a Gaussian mixture detailed in section 3.4.

We believe that we did not put forward this reasoning clearly at first, leading to the general comment that our experiments did not enough support our claims. In the new draft however, thanks to your help, we believe this confusion to be no longer present. We also included additional experimental results to validate our neuroimaging experiment.

---

> ### Author Response · Authors · 2021-11-20
> **2/2**
>
> ## Detailed Changelog
>
> * **Introduction**:
>     * we engage with the general framing of approximate inference, and notably with the notion of the *approximation vs amortization gap*.
>     * our main claim has been made clearer: we propose a method to benefit from the expressivity of normalizing flows in the context of population studies featuring a large number of latent parameters
> * **Methods**:
>     * we better motivate the class of pyramidal models, the utility of this class of problems, and how it can lead to a large number of parameters, as seen in section 2.1
>     * we restate our main claim in a conclusive manner, in section 2.3
> * **Experiments**:
>     * we reworked the presentation of Table 1 and Figure 3 to put forward more clearly the role of MF-VI as a proxy to the ground truth posterior
>     * we included multiple paragraphs dedicated to the explanation of the reasoning stated in our *General comment* section above, for instance in section 3.2
>     * we restated our discussion using the framework of the approximation vs amortization gap, for instance in section 3.3
>     * we removed the ambiguity between different hardware -focusing on CPU process time- as well as in the comparison between amortized versus non amortized techniques -separating the reporting of the amortization vs the inference time- notably in Table 1
>     * we slightly reworked Figure 2a
>     * we completely reworked our neuroimaging section 3.5, adding a quantitative comparison with MF-VI, building up an experiment showing the practical interest there is in amortization, and explicitely stating that we are the only benchmarked flow-based method able to infer in this parameter regime
> * **Discussion**:
>     * we removed some references to likelihood-free methods, and instead engaged more generally with approximate inference
>     * we moved our discussion about the Mean Field approximation from our supplemental material to our main text
> * **Supplemental material**:
>     * we added supplemental discussion related to:
>         * the relevance of likelihood-free methods in the presence of a likelihood (B.3)
>         * the inference over a subset of the latent parameters (B.4)
>         * the embedding size for the Hierarchical Encoder (B.5)
>         * upper bounds for ADAVI's inference performance (B.6)
>     * we reworked supplemental Table 1, that represents the results from Fig. 3 (main text), according to the changes made in our *Experiments* section
>     * we moved the equation of the MSHBM (neuroimaging model) -previously equation 9- to section E.1
>     * we added a new section E.5.2 and a new figure E.7 related to inference over subsets of the population, and showing the interest in amortization, as referenced to in section 3.5 (main text)
> * **General**: we addressed comments related to typos and unclear texting. Thank you for your vigilance.
>
> We carefully ensured that your comments were individually addressed in this revision. We stand at your disposal shouldn't this be the case.
>
> We thank you again for your time, and your help in improving our work,
> Best regards,
> The authors

---

> > ### Comment · Reviewer_bbMw · 2021-11-21
> > **I like the new revision but I still don't get the MF-VI comparisons**
> >
> > I still don't get the argument that the proposed methods can outperform MF-VI as stated, and I don't think conjugacy matters for this point (clearly MF-VI is both possible and common in the non-conjugate case, in cases as simple as Bayesian logistic regression). I also don't think that MF-VI needs to be a proxy for the ground-truth posterior to be a relevant upper-bound for amortization methods.
> >
> > As I see it: in MF-VI you have a $q(\theta \mid X) \approx \prod_i q(\theta_i)$ where each datapoint gets to have its own parameter $\theta_i$ independent of the data. In an amortization setup, each datapoint no longer gets to have its own parameter $\theta_i$ and rather needs to be approximated as a function of the data so you have $q(\theta \mid X_i)$. As long as you haven't changed the approximating family between the MF-VI setup and the amortized setup, the best that the amortized setup can do is the same separate-parameter-per-datapoint approximation as MF-VI. On the other hand, if the approximating family is different between MF-VI and an amortized method, it's not clear what to make of the comparison. Am I missing something here? I see that reviewer bt4m shares the same perspective regarding limiting behavior approaching MF-VI. At this point it just seems technically incorrect, and while the overall paper is improved I'm really concerned about having a paper with potentially incorrect understanding of VI be published at a major conference.
> >
> > Also, a minor comment: section 3.1 new sentence "In terms of ELBO, on top of an approximation gap an amortized technique will addition an approximation gap" -> probably meant to say "amortization gap" in the latter case? And possibly something like "will add" or "will additionally include" rather than "will addition".

---

> > > ### Author Response · Authors · 2021-11-22
> > > **We argue that MF-VI as defined in our work is not a limit case for ADAVI, and derive an alternative limit case detailed in our supplemental discussion.**
> > >
> > > Dear reviewer,
> > >
> > > Thank you for your answer and your engagement in this rebuttal process. We understand your concern about correctness. We propose to detail our reasoning in several points:
> > >
> > > * MF-VI is defined in our work -following common practice (Blei et al. 2017, Bishop 2006)- as the product of independent densities with the **same parametric form as the prior**, as seen in section F;
> > > * With that definition, MF-VI is **not** a limit case for ADAVI, and can suffer from poor performance in non-conjugate cases, as seen in section 3.2;
> > > * A mean field distribution using a product of **independent normalizing flows** -not present in the literature as far as we know- would in a sense a limit case for ADAVI. This limit case wouldn't be much more adapted to large population studies than Cascading Flows (Ambrogioni et al. 2021) -its parameterization would similarly scale linearly with the plate cardinalities;
> > > * ADAVI differs from that limit case by 3 aspects: **shared parameterization, amortization, and automation** (section 2.2). The first 2 points entail a potential gap in inference quality;
> > > * Contrary to what a "limit case" would entail, those 3 differentiating points are not detrimental as they result in:
> > >     * our plate cardinalities-invariant parameterization -that allows us to tackle **data regimes unreachable for existing flow-based methods**, as in section 3.5;
> > >     * the breaking of some of the barriers to entry for general experimenters to perform large-scale VI, as presented in our introduction and discussion;
> > >
> > > You can see more details on this topic in our **supplemental section B.6, uploaded as part of a new rebuttal draft**.
> > >
> > > As a concluding note, we would underline that **the Mean Field approximation** is a readily modifiable implementation detail, and **is not tied to our main claim**, as underlined in our discussion.
> > >
> > > Thank you again for your time,
> > >
> > > Best regards,
> > >
> > > The authors

---

### Decision · Program_Chairs · 2022-01-20

**Decision:**

Accept (Poster)

**Comment:**

The paper provides a unique contribution to the scalability of Bayesian inference to Pyramidal Bayesian Models with application to neuroimaging. The major point of concern by the reviewers is around how close is the inference approach to the more classical Mean-Field VI. However, in my opinion, the authors have addressed these concerns in the rebuttal. Therefore, I recommend Accept.